# TRIVIAL OR IMPOSSIBLE—DICHOTOMOUS DATA DIFFICULTY MASKS MODEL DIFFERENCES (ON IMAGENET AND BEYOND)

**Kristof Meding**[*]
University of Tübingen
`kristof.meding@uni-tuebingen.de`

**Luca M. Schulze Buschoff**[+*]
University of Tübingen
`luca.schulze-buschoff@student.uni-tuebingen.de`

**Robert Geirhos**
University of Tübingen & IMPRS-IS

**Felix A. Wichmann**
University of Tübingen

## ABSTRACT

*"The power of a generalization system follows directly from its biases"* (Mitchell 1980). Today, CNNs are incredibly powerful generalisation systems—but to what degree have we understood how their inductive bias influences model decisions? We here attempt to disentangle the various aspects that determine how a model decides. In particular, we ask: what makes one model decide differently from another? In a meticulously controlled setting, we find that (1.) irrespective of the network architecture or objective (e.g. self-supervised, semi-supervised, vision transformers, recurrent models) all models end up making similar decisions. (2.) To understand these findings, we analysed model decisions on the ImageNet validation set from epoch to epoch and image by image. We find that the ImageNet validation set, among others, suffers from dichotomous data difficulty (DDD): For the range of investigated models and their accuracies, it is dominated by 46.0% "trivial" and 11.5% "impossible" images (beyond label errors). Only 42.5% of the images could possibly be responsible for the differences between two models' decision boundaries. (3.) Only removing the "impossible" and "trivial" images allows us to see pronounced differences between models. (4.) Humans are highly accurate at predicting which images are "trivial" and "impossible" for CNNs (81.4%). This implies that in future comparisons of brains, machines and behaviour, much may be gained from investigating the decisive role of images and the distribution of their difficulties.

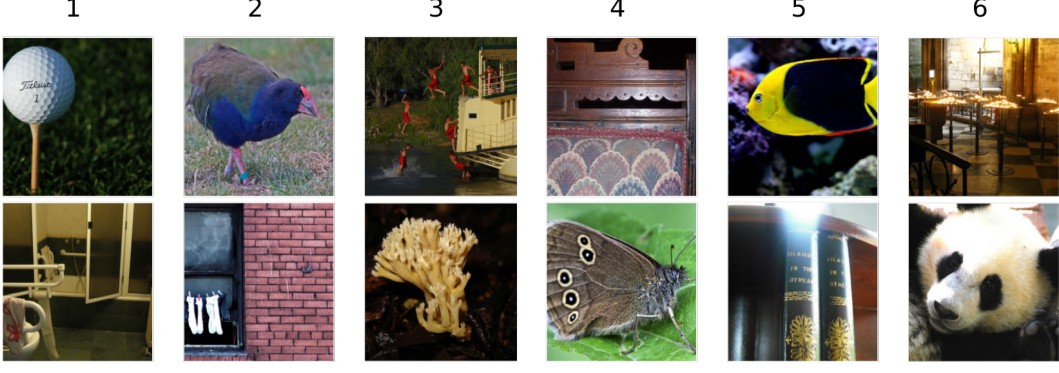

Figure 1: Can you predict which of these images are "tricky" for CNNs? Out of every of the six pairs, one image is always correctly classified and one always incorrectly (answers on the next page[1]). On ImageNet, image difficulty appears largely dichotomous: CNNs make highly systematic errors irrespective of inductive bias (architecture, optimiser, ...). Humans can reliably differentiate between images that are "trivially easy" and "impossibly hard" for CNNs (81.4% accuracy).

---

[*]joint first authors in alphabetical order; [+]corresponding author

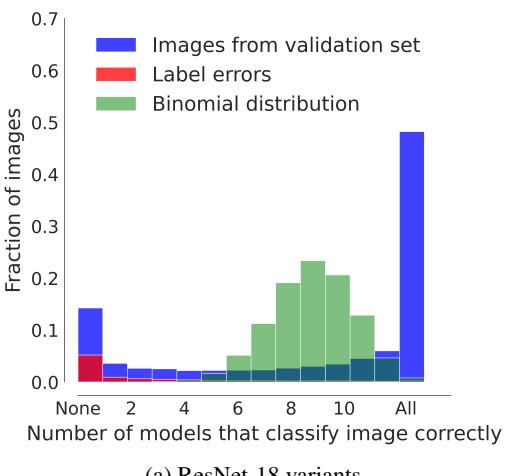
(a) ResNet-18 variants

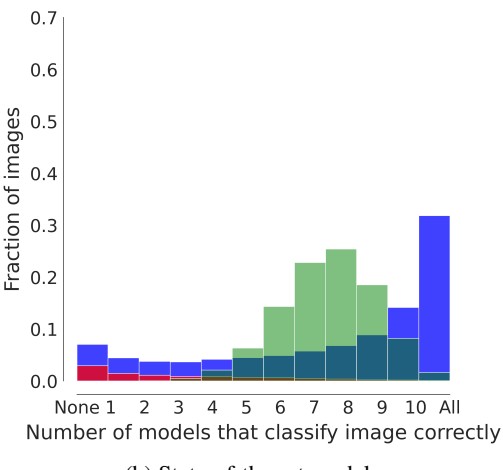
(b) State-of-the-art models

Figure 2: Dichotomous Data Difficulty (DDD) in a nutshell: Irrespective of model differences (e.g. architecture, hyperparameters, optimizer), most ImageNet validation images are either "trivial" (in the sense that all models classify them correctly) or "impossible" (all models make an error). This dichotomous difficulty masks underlying differences between models (as we will show later), and it affects the majority of the ImageNet dataset—i.e. not only images with label errors (red) as identified by the cleanlab package (Northcutt et al., 2021a). For comparison, a binomial distribution of errors is shown in green: this is the distribution of errors expected for completely independent models if all images were equally difficult.

# 1 INTRODUCTION

Let's play a game we call *Find those tricky images!* In Figure 1, we show pairs of images. One image is impossible for a CNN regardless of its architecture, optimiser, random seed etc.—it never gets the label correct. The other image always yields a correct classification—can you find the tricky images?

Done? We will wait. You have probably never seen these images before, and neither have CNNs seen them during training. How exactly a decision maker—be it a neural network, or a biological brain—generalises to previously unseen images is influenced by the decision maker's *inductive bias* (Goyal and Bengio, 2020)—in fact, as already recognised in 1980, "the power of a generalisation system follows directly from its biases" (Mitchell, 1980). Commonly, the inductive bias is defined as the set of assumptions and choices that determine which hypothesis space is available to the model, before the model is exposed to data. For instance, starting from the set of all possible hypotheses, the hypothesis space of linear models is a tiny subset (linearity is one example of a strong inductive bias). After the "choice" of the inductive bias, the dataset then influences which particular decision boundary (or concrete hypothesis) is selected from the model's hypothesis space. Finding the right inductive bias for a given problem is at the core of machine learning. Therefore it is only consequent that a tremendous amount of work is being invested in improved architectures (Alzubaidi et al., 2021), optimisers (Ruder, 2016), learning rate schedules(Loshchilov and Hutter, 2016), etc.—surely we would expect these choices to make a difference on the resulting model's decisions even if trained on the exact same dataset. However, in the present work, we have tested various factors related to the inductive bias—among other aspects, architecture, optimiser, learning rate, and initialisation—and yet, on ImageNet, all models agree in the sense that they *all* make largely similar errors. This is shown in Figure 2: even radically different state-of-the-art (SOTA) models make surprisingly similar errors on the ImageNet validation set. To a certain degree, image difficulty appears dichotomous: nearly 60% of all images are either "trivial" (all models correct) or "impossible" (all models wrong). As we will demonstrate later, this dataset issue masks and overshadows hidden differences between models.

---

[1]The tricky (=misclassified) images are: 1. bottom, 2. bottom, 3. top, 4. top, 5. bottom, 6. top. This game is an homage to "Name that dataset" by Torralba and Efros (2011).

## 1.1 RELATED WORK

**Metrics for CNN comparisons**   Given the scientific, practical and engineering implications of model inductive biases, it is perhaps not surprising that many studies investigated differences between neural networks. For this purpose, the standard metric is accuracy, but some studies also focus on learned features and decision boundaries (e.g. Hermann and Lampinen, 2020; Nguyen et al., 2020; Wang et al., 2018; Hermann et al., 2019; Shah et al., 2020), or internal representations (Kriegeskorte et al., 2008; Kornblith et al., 2019). Using representational similarity analysis (RSA) and most similar to our work, Mehrer et al. (2020) and Akbarinia and Gegenfurtner (2019) investigated whether different CNNs yield correlated representations and found that many neural networks show differences on a representational level. How intermediate representations are related to classification behaviour largely remains unclear. In order to compare networks on a behavioural level directly, metrics such as *error consistency* can be used. Error consistency (measured by $\kappa$) assesses the degree of agreement between two decision-makers on an image-by-image basis, not just average performance (Geirhos et al., 2020a;b).

**Consistent model errors**   Tramèr et al. (2017) observe that the decision boundaries of two models are highly similar, an issue that is related to the transferability of adversarial examples between models. Additionally, it has been shown that standard vanilla models systematically agree on their errors both on IID (independent and identically distributed) data (Mania et al., 2019) and OOD (out-of-distribution) data (Geirhos et al., 2020a). It is unclear whether, if at all, there is a connection between model inductive bias, dataset difficulty and consistent model errors. Another line of work investigated fairness metric consistency (Qian et al., 2021).

**Relationship between DDD and OOD**   Evaluating models on out-of-distribution (OOD) datasets is an important way to differentiate between models. However, different models show highly consistent errors even when evaluated on OOD data, according to Geirhos et al. (2020a, Figure 3). Here, CNN-to-CNN consistency is at .62, .48 and .67 depending on the OOD dataset, which is closer to perfect consistency than it is to chance level. Therefore, while OOD data can distinguish models in terms of overall accuracy, OOD testing is insufficient in terms of revealing deeper differences overshadowed by dichotomous data difficulty (DDD). Furthermore, OOD datasets might also exhibit DDD. Looking forward, OOD testing as well as curating datasets without DDD are not mutually exclusive and should be employed in a combined fashion for a comprehensive understanding of model similarities and differences.

**Problems of datasets**   The ImageNet dataset (Russakovsky et al., 2015) has numerous issues, including some that affect most datasets, like dataset bias (Torralba and Efros, 2011). Northcutt et al. (2021b) showed that around 6% of ImageNet validation images suffer from label errors. Additionally, many images require more than a single label since multiple objects are present, and the distinctions between classes seem rather arbitrary at times (Tsipras et al., 2020; Beyer et al., 2020). Even when trying to replicate the original ImageNet labeling procedure in order to create a new test set, models trained on ImageNet have an accuracy drop of 11–14% on this new test set (Recht et al., 2019). Finally, ImageNet labels are based on the WordNet hierarchy, which contains many problematic categories. For instance, many categories in the "person" subtree have labels ranging from outdated to outrageous and racist (Crawford and Paglen; Yang et al., 2020). Furthermore and similar to our work, authors already investigated image sampling strategies during training (Jiang et al., 2019; Katharopoulos and Fleuret, 2018). However, these studies focused on accelerating the training and not how the ImageNet issues may obscure differences between models as we explore here.

**Example difficulty**   A number of papers have investigated what makes images easy or difficult—e.g. Agarwal et al. (2020); Baldock et al. (2021); Mangalam and Prabhu (2019); Paul et al. (2021) for MNIST/CIFAR, and e.g. Hacohen et al. (2020) for ImageNet. Additionally, it is well-known that models often make similar errors and often learn examples in the same order, see e.g. Toneva et al. (2018); Kalimeris et al. (2019).

To summarise: it was clear that there are easier and harder images and that models often make similar errors. However, the relationship between these two findings has not commonly been recognised. We here show for the first time the implication thereof: That underlying model differences are masked by dichotomous data difficulty.

## 2 METHODS

All details regarding our software, hardware and dataset can be found in the appendix (section A.2).
**Similarity measure** For the investigation of network similarities, we mainly use the behavioural measure error consistency ($\kappa$) (Geirhos et al., 2020a) based on Cohen's work (Cohen, 1960). $\kappa > 0$ indicates that two decision-makers systematically make errors on the same images; $\kappa = 0$ indicates no more error overlap than what could be expected by chance alone. $\kappa < 0$ implies that two decision-makers systematically disagree.

**Network variations** In our experiments we investigated the systematic agreement between CNNs, varying not only architecture but carefully controlling for number of epochs, optimiser, batch size, random initialisation, learning rate, hardware randomness, data order, architecture, and disjoint data sampling. Unless stated otherwise, we only changed one of the above parameters at a time. Our main results are based on the ImageNet ILSVRC dataset (Russakovsky et al., 2015). We first used systematic variations on ResNet-18 (called ResNet-18 variants). Details can be found in section A.1 in the Appendix. In total, 30 networks were trained on each of the three data sets (See below: ImageNet, CIFAR-100, Gaussian) presented in the main text, as well as 60 more networks for control experiments reported in the Appendix. We stored all network states and all responses for each epoch. This allows us to analyse the agreement on different training stages epoch by epoch (and image by image). Later, we investigated different state-of-the-art network architectures. When we investigated these SOTA models, implementations provided by `modelvshuman` (Geirhos et al., 2021) were used (which focuses on various out-of-distribution datasets but not on ImageNet as we do).

**Psychophysical experiment** We conducted two psychophysical experiments. For both experiments detailed methods can be found in section A.10 in the appendix.

## 3 RESULTS

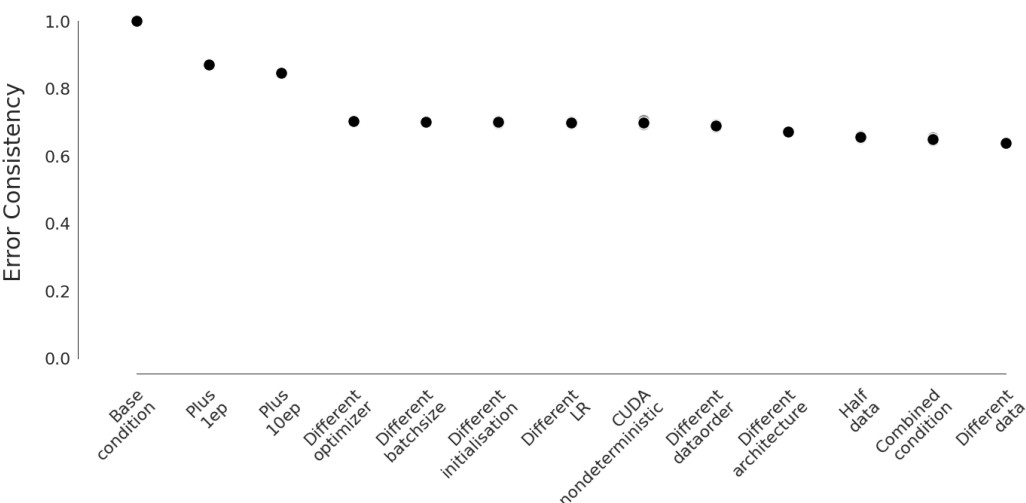

Figure 3: Error consistencies between the different conditions and the base network on the ImageNet validation set after 90 epochs. For conditions for which multiple models were trained the mean over all models of a condition is plotted in black.

### 3.1 MODEL ERRORS ARE ALIGNED DUE TO DICHOTOMOUS DATA DIFFICULTY (DDD)

Figure 3 shows the result of our controlled study of model differences on ImageNet. A positive error consistency score means that the networks agree beyond what is expected by independent models. Regardless of the parameter changed (architecture, optimiser, etc.), we find very high error consistencies (around 0.7)—thus all models agree which images are easy or difficult to classify irrespective of the model differences investigated.[2] Strikingly, changes that we hypothesized would

---

[2]In addition to the mean across several runs, we also plot the consistencies of single runs in gray. However, these are non-visible since the variance within the conditions is very small (except, perhaps, for the "cuda nondeterministic condition").

make a larger difference, e.g. different architecture, have basically the same error consistency as "minor" changes like enabling hardware randomness on the GPUs. All networks achieved similar top-1 accuracies (mean: 69.05% after 90 epochs; range: 65.87% to 71.47%; standard deviation: 1.60%, cf. Figure 13 in the Appendix). Another popular method for agreement analysis is RSA. All our results also hold here, see Figure 7 in the Appendix. Additionally, switching the base architecture to VGG-11 or DenseNet-121 does not make a difference either (see Figures 15 and 16 in the Appendix). A deeper analysis of a single network's decisions can be found in section A.5 in the appendix.

The findings from Figure 3 becomes even more prominent in Figure 4 where we overlay the previous figure for all of the 13 networks with different hyperparameters, architectures etc. (explained in section 2). A very light red entry indicates that *all* networks correctly classify the image, a very dark red entry that all networks misclassify the corresponding image; shades of red indicate the cases in-between (where, e.g., some but not all networks make errors). The figure illustrates that the previous findings *hold across very different inductive biases for ResNet-18 variants*: We observe that 48.2 % images are learned by all models regardless of their inductive bias; 14.3 % images are consistently misclassified by all models[3]; only roughly a third (37.5%) of images are responsible for the differences between two model's decisions. We call this phenomenon dichotomous data difficulty (DDD): While the inductive bias restricts the hyperparameter space for a given model, the nature of the dataset—and especially its highly non-uniform image difficulties—seems to be an important cause for the high similarity in the decisions of different networks. Model differences may play a bigger role for images of intermediate difficulty—where there is substantial consistency variation across models—but only a minor role for easy and hard images. As the dataset primarily consists of images that all models either classify correctly or incorrectly, all models end up with similar classification behaviour.

Let us consider two extreme cases in order to put these findings into context. On one end of the spectrum, if all images were equally difficult *and* if all networks were independent (i.e. their different inductive biases would result in independent decision boundaries), then we could expect a binomial distribution of model errors: out of 13 investigated ResNet-18 models, very few images should be misclassified by all models and very few correctly classified by all models—instead, most images should be correctly classified by a handful of models. Figure 2a shows, in green, exactly this distribution expected for independent models and equally difficult data [4]. On the extreme end of the spectrum, if the inductive bias had no influence at all and the dataset only contained "trivial" and "impossible" images, we would expect a histogram with only two "spikes": given ImageNet accuracies of 69.05% on average, one spike at "None" (30.95% for ImageNet) and one at "All" (69.05 % for ImageNet). Clearly, the empirically obtained histogram (blue) much more resembles the latter, i.e. the scenario where the (nearly) dichotomous data difficulty dominates over inductive bias. We observe that DDD on ImageNet is amplified, but not caused, by label errors (Northcutt et al., 2021a; Beyer et al., 2020; Tsipras et al., 2020) which only have a minor influence on the "None-Bar" from our histogram in Figure 2. Hence: removing erroneous labels is beneficial and laudable, but it will not solve DDD.

Is dichotomous data difficulty (DDD) only a problem for ImageNet? This is not the case: DDD is also present in CIFAR-100 and in the synthetic Gaussian dataset we (purposefully) generated . As a first indication, for both of these datasets we find similarly high error consistencies between different models, just like we found for ImageNet (see section A.6 in the appendix).

## 3.2 DICHOTOMOUS DATA DIFFICULTY EVEN AFFECTS RADICALLY DIFFERENT STATE-OF-THE-ART MODELS

In the previous section, we found that changing different aspects within *one model class* does not change the decisions significantly. However, it is unclear whether these results generalize across model classes. Therefore, we apply the analysis from Figure 2a with a number of models specifically chosen to be radically different from each: a self-supervised model (SimCLR, Chen et al. (2020)), a semi-supervised model (SWSL, Yalniz et al. (2019)), a vision transformer (ViT, Dosovitskiy et al. (2020)), a recurrent model (CORnet-RT, Kubilius et al. (2019)), a very deep model (ResNet-152,

---

[3]Of course, these numbers change if one uses an architecture with higher top-1 accuracy, see next section

[4]As one of our reviewers suggested, we also tested a model in which the image difficulty is decaying exponentially—most images are simple, then less and less are more difficult. However, this also does not lead to a DDD like distribution (see A.9).

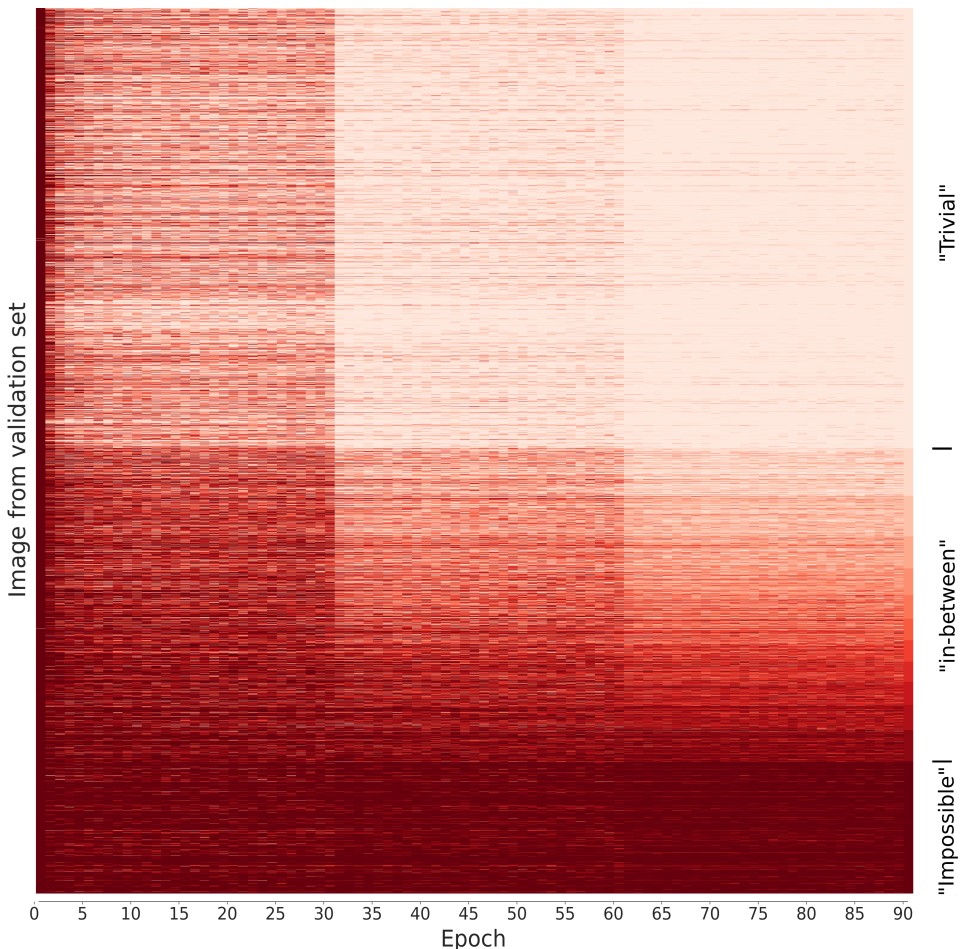

Figure 4: Decisions on all 50K ImageNet validation images of *all* 13 networks with different inductive biases (architectures, ...). Dark red indicates that the respective item was falsely classified by all networks. Light red indicates that the image was correctly classified by all networks. Images are ordered according to the mean accuracy across networks in the last epoch.

He et al. (2016)), a highly compressed model (SqueezeNet, Iandola et al. (2016)), an adversarially trained model (ResNet-50 with epsilon 1 L2-robustness on ImageNet, Salman et al. (2020)), a bag-of-local-features model (Bagnet-33, Brendel and Bethge (2019)), a network trained on stylized ImageNet (ResNet-50 trained on SIN, Geirhos et al. (2019)), a deep high resolution neural network (HRNET, Wang et al. (2020)), and OpenAI's CLIP model (Radford et al., 2021) with a transformer architecture and joint image-language training objective (11 models in total). Individual accuracies of these networks can be found in the appendix (see Figure 14). Again, we find the same pattern in Figure 2b. In total, 46.0 % "trivial" images are learned by all except one model; 11.5 % "impossible" images are consistently misclassified by all except one model. (42.5%) of images are responsible for the differences between two model's decisions.

## 3.3 Dataset subsampling according to Dichotomous Data Difficulty reveals differences between models

So far we have seen that models agree despite markedly different choices of architecture, training objectives, and many other aspects. While we hypothesized DDD—a dataset problem—to be the cause, an alternative explanation would be that models simply agree irrespective of the choice of data difficulty. In order to differentiate between these two competing hypotheses we performed an experiment where we removed both the "trivial" and the "impossible" images from the ImageNet

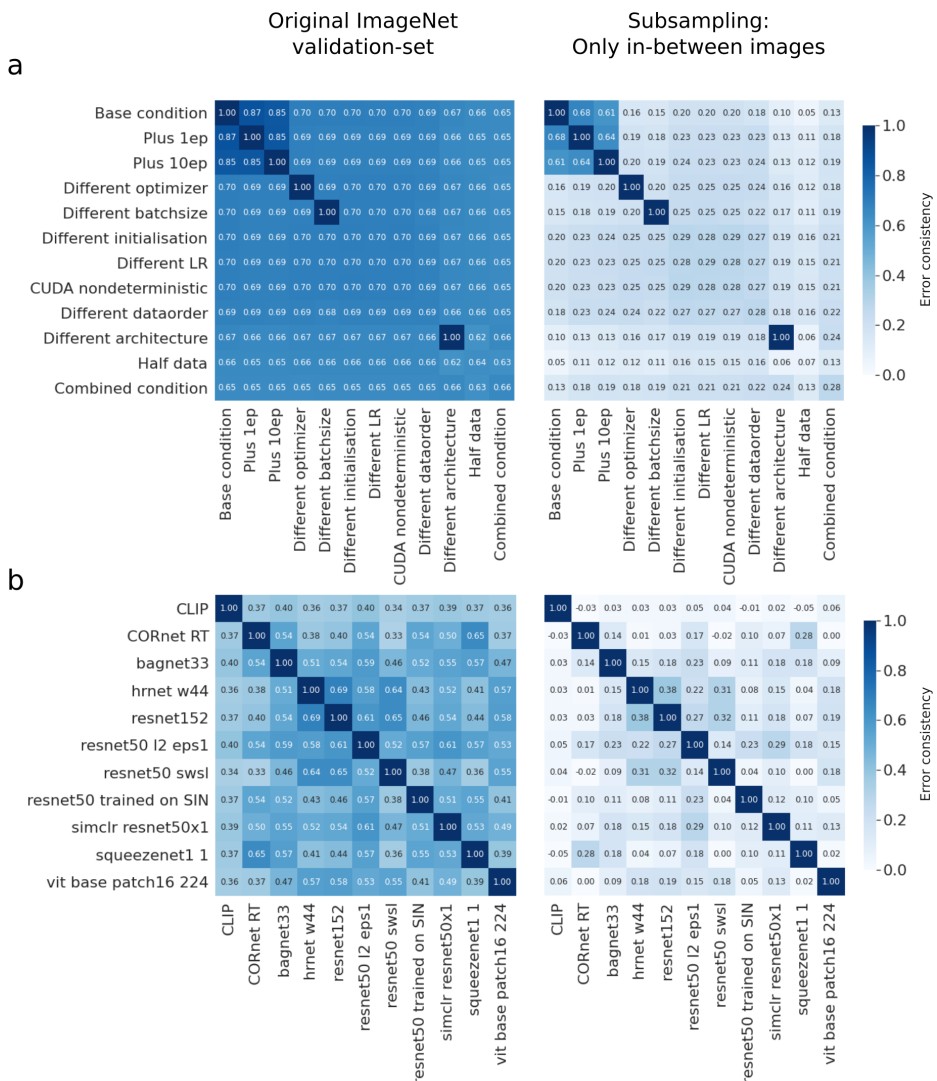

Figure 5: Error consistency on the original ImageNet test-set (left panel) and on the test-set with in-between images only (right panel) for the ResNet-variants (a) and the SOTA networks (b). Both networks were trained on the whole ImageNet training set. Error consistency around 0 indicates independent responses. A diagonal element of 1 represents that only one network for comparison was available, otherwise the within condition consistency is calculated, see section 2.

validation dataset. The training dataset was not altered. If model agreement is indeed caused by DDD, then we should find much stronger differences between different models (as indicated through lower error consistency scores). The results are presented in Figure 5: Indeed, model differences are now much more pronounced, in many cases the consistency between different models even approaches zero, indicating that some networks make truly independent decisions, i.e. have learned independent decision boundaries whilst being similarly accurate—their different inductive biases now show. This shows that the high agreement between different models (as observed e.g. by Geirhos et al. (2020a; 2021) and Mania et al. (2019)) is a result of dataset DDD problems, not that inductive bias does not matter much. Please note that the reduced consistency is not trivially caused by the removal of impossible and trivial images: Even when removing extreme images (all models correct/incorrect), two models could agree or disagree on the remaining images of intermediate difficulty (error consistency is calculated pair-wise). Finally, we show that there are some particularly easy and hard classes (Section A.8 in the Appendix).

### 3.4 Differences between "trivials", "impossibles" and "in-betweens"

Since we found DDD to affect very different models, we were interested to understand the nature of their differences. To this end, we asked in our first experiment whether humans could identify which images were "trivial" and "impossible" for CNNs. If they can, this would mean that there is—at least to some degree—a shared notion of image difficulty between humans and CNNs. We therefore conducted a psychophysical experiment where subjects were asked to identify which of two images was easier for a neural network to classify. We found that human observers were able to do so well beyond chance (50%): on average, with an accuracy of 81%. The accuracies of the different subjects ranged from 72%[5] to 89%, with a standard deviation of 6.29%. The mean error consistency between the subjects was 0.59. For all combinations of different subjects, the error consistency ranged from 0.41 to 0.75, with a standard deviation of 0.09. In conclusion, even naïve human observers without machine learning experience can reliably and consistently predict which images are easy and difficult for CNNs. In the Appendix, we also show that the ImageNet "superclasses" are not equally distributed for the three image subsets (see section A.12) and we observe that label ambiguity does not seem to be the major cause for the emergence of "in-betweens" (see section A.13).

Additionally, we conducted a second follow-up human experiment which aims to find decisive factors for the trivial, in-between and impossible images. We randomly chose 100 "trivial", "impossible" and "in-between" images, each of our eight human observers had to answer three questions; Q1: "How many objects belonging to different categories are in the image?', Q2: "Is there a main category in the image?" Q3: "Is the presentation of the objects unusual in any manner?". The main results are summarised in Figure 6.

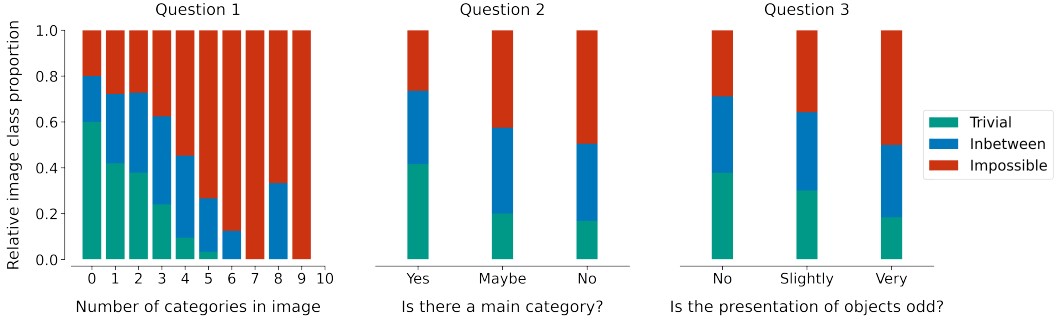

Figure 6: Barplot displaying the proportions of answers over all observers and for the three questions mentioned below the subfigures. Colour coded is the image subclass.

For this figure, we removed images which were found to have label errors by Northcutt et al. (2021b) and balanced the image classes to have the same number of images (n = 57 per class, otherwise our plots would be misleading if we do not have the same number of images per class). The bars are normalized so that the proportions of the different classes add up to 1 for each answer category. We see a clear trend, that with increasing number of categories, increasing uncertainty about the presence of a main category and increasing oddity of presentation, the proportion of "trivials" decreases while the proportion of "impossibles" increases. The Pearson correlation coefficient between the mean number of categories over all observers for each image and the respective mean model accuracy is -0.37. Furthermore, we found that items with a clear main category had a mean accuracy of 0.64. Items where observers indicated that they "maybe" had a main category had a mean accuracy of 0.44 while items with no main category had a mean accuracy of 0.37. Finally, items with a normal presentation had a mean model accuracy of 0.61. Items with a slightly odd presentation had a mean model accuracy of 0.55 and items with a very odd presentation had a mean model accuracy of 0.34. We also find high consistencies between the observers, see section A.11 in the appendix. For completeness, we show raw pooled data in Figure 26/25 (with/without label errors) as well as individual data with and without label errors in Figure 27/28. All our results also hold when we include images with label errors.

---

[5]Our experimental paradigm here is a high-powered small N-Design (Smith and Little, 2018). Even our worst performing observer with 72% accuracy yields a p-value of $5 \cdot 10^{-8}$ for the null hypothesis of chance performance.

In summary, experiment one shows that human can reliably and with high accuracy distinguish trivial and impossible image. Second, we found in our second experiment that the number of categories in the image, whether there is a main category or not, and the oddness of presentation all seem to contribute to whether an image belongs to the "trivial", "in-between" or "impossible" subsets. While the factors we investigated show a clear effect, it is evident that there is no single factor that completely explains the differences between the three subsets of images. We are clearly still in need of an explanation what makes images "trivial", "in-between" or "impossible" for neural networks. To a certain degree, this is perhaps not surprising. There is a long history of investigating the factors underlying image difficulty in vision research which we could not capture with our only three questions. Future studies might draw on this line of research and ideally explore more possible factors which could be related to the dichotomous difficulty inherent in popular image datasets.

## 4 DISCUSSION

We investigated the influence of dataset difficulty on model decisions. We found that model decisions are not only determined by the inductive bias (such as their architecture)—they are even more influenced by the dichotomous difficulty of images in common datasets (DDD): many ImageNet images are either "trivial" or "impossible", but only a third in-between. This has implications for model design. Viewed positively, results for one network may generalise towards different networks, which can be advantageous in some circumstances. This is in line with previous findings that some results transfer between different model classes, e.g. adversarial examples (Szegedy et al., 2013; Papernot et al., 2016). However, if models are trained on datasets with DDD, design decisions like architectural improvements may not be able to show their full potential since the resulting models, due to DDD, have a high likelihood of ending up in a very similar decision-regime as other (already existing) models—and might even inherit their vulnerabilities. In comparison to underspecification described by D'Amour et al. (2020) we observe that models behave very similarly because of DDD. When removing trivial and impossible images, the differences between models are unmasked—which potentially can be used to accelerate training (Jiang et al., 2019; Katharopoulos and Fleuret, 2018). Furthermore, our method also offers an easy to implement method to curate DDD-free datasets. One only needs to train different models on the same data, followed by removing impossible and trivial images.

Previous investigations found label errors to be a problem in a number of datasets. Here we show a dataset issues that affects a much larger number of ImageNet images than those affected by label errors. In order to be able to improve our ability to differentiate between models and give their inductive bias a chance to truly make a difference, we will need datasets that are more balanced with respect to image difficulty or use only in-between images. This is far from trivial since we do not know precisely what causes DDD. Inspiration could come from psychology and vision science, where investigations into what makes an image or object difficult have a long history. At least since Eleanor Rosch's (Rosch, 1973) pioneering work we know that for some object categories there are "natural prototypes", i.e. particularly representative exemplars of a category. Thus not all members of a category are equally easy to recognize and classify. Second, for human vision it is well known that the recognition of an object depends on its viewpoint: objects are easier to see from a "canonical" viewpoint (Biederman, 1987; Bülthoff and Edelman, 1992; Freeman, 1994; Tarr and Kriegman, 2001; Tarr et al., 1996). Third, object recognition also depends on its context and surroundings. Humans can recognize objects remarkably quickly (Thorpe et al., 1996), but this is only true if they are effectively segmented from their background by the photographer's selection of focus point, focal length and aperture (Wichmann et al., 2010). As a result one can make a real-world dataset arbitrarily trivial (or impossible) for human observers. Perhaps it was somewhat naïve to believe that large automatically generated datasets would "get the mix right" and result in images where the difficulty within and between categories is approximately the same.

Our human experiment shows that humans can reliably identify the impossible images from ImageNet (see Figure 18 in the Appendix for more examples). Inspection of those images left us with the impression that impossible images often contain multiple objects and sometimes "unusual" objects and viewpoints which is verified in our follow-up experiment. From a cognitive science or neuroscience perspective DDD might thus also provide new opportunities for insight: Perhaps the impossible images are the ones which can reveal differences between humans and CNNs and are thus those which neuroscience and cognitive science should be interested in?

## 5 ETHICS STATEMENT

**Potential social harm.** We do not expect that our work causes harm to people or groups.
**Environmental aspects.** We roughly used 250 GPU days for this paper. Each GPU unit on our cluster (together with CPU and RAM) consumes on average 300W. In total, this paper consumed 1800kWh. The $CO_2$ emission in the country of the authors is roughly 400g/kW resulting in a $CO_2$ equivalent of 720kg—this corresponds to roughly 45% of the emission of a flight from London to New York. We will compensate the amount of $CO_2$ with a certified $CO_2$-compensation company. Furthermore, we will make sure that other researchers have access to the trained models, see below. We can not distribute all models yet (several GBs) because of the size limit of the supplementary materials.
**Psychophysical experiment.** Prior to the experiment written consensus was collected from all participants. Recently, some issues around ImageNet were discussed e.g. by `https://www.excavating.ai`. Thus, we removed some images in our psychophysical experiment and do not show any images containing humans in this paper. Otherwise, we do not see potential participant risks in our experiment.

## 6 REPRODUCIBILITY STATEMENT

Code to reproduce our findings can be found on github: `https://github.com/wichmann-lab/trivial-or-impossible`.

### ACKNOWLEDGMENTS & FUNDING DISCLOSURE

Funding was provided, in part, by the Deutsche Forschungsgemeinschaft (DFG, German Research Foundation)—project number 276693517—SFB 1233, TP 4 Causal inference strategies in human vision (L.S.B, K.M. and F.A.W.). The authors thank the International Max Planck Research School for Intelligent Systems (IMPRS-IS) for supporting R.G.; and the German Research Foundation through the Cluster of Excellence "Machine Learning—New Perspectives for Science", EXC 2064/1, project number 390727645, for supporting F.A.W. The authors declare no competing interests.

We would like to thank Silke Gramer, Leila Masri and Sara Sorce for administrative and the "Cloudmasters" of the ML-Cluster at University Tübingen for technical support. We thank the group of Ludwig Schmidt at UC Berkeley for discussions during early stages of this project and David-Elias Künstle for helpful comments on the manuscript. This work benefited considerably from conscientious peer-review: We would like to express our sincere gratitude to all our reviewers.

### AUTHOR CONTRIBUTIONS

Motivated by previous work (Geirhos et al., 2020a), the project was initialized by K.M. and jointly developed forward with R.G. and F.A.W. Later, but still at an early stage, L.S.B. joined the project. L.S.B. wrote the code for computing and analysis and was supervised by K.M.; R.G .and F.A.W. gave input during the entire project. R.G. and F.A.W. had the idea of the psychophysical experiment. All authors planned, structured and wrote the manuscript; all figures were made by L.S.B. based on joint discussion.

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

# A    APPENDIX

## A.1    RESNET-18 VARIANTS

Our systematic variations for the ResNet-18 variants are:

- *Base condition*: a standard ResNet-18 trained on ImageNet in PyTorch[6] was used as the baseline network for all comparisons; one instance trained.
- *Plus 1ep*: a network was trained for one additional epoch compared to the base network; one instance trained.
- *Plus 10ep*: a network was trained for ten additional epochs compared to the base network; one instance trained.
- *Different optimizer*: a network was trained using SGD with Nesterov momentum(Sutskever et al., 2013) instead of vanilla SGD; one instance trained.
- *Different batch size*: for this condition we split the batch size in half (128 instead of the 256). This was done by drawing the same batches from the data loader and splitting them in half. We then input the halves sequentially into the model, effectively doubling the number of gradient updates; one instance trained.
- *Different initialisation*: networks were varied in the initialisation of their layer weights by choosing a different random seed for each network; five instances trained.
- *Different learning rate*: the networks were trained using initial learning rates varying from 0.148 to 0.152 instead of the default learning rate of 0.1. We narrowed the range such that they still reach the same accuracy level; five instances trained.
- *CUDA non-deterministic*: Training networks without CUDA determinism is the standard procedure. However, graphic card operations are not necessarily deterministic, e.g. functions like `reduce_sum` (Riach, 2019). This non determinism might not influence accuracy but may influence agreement between instances; five instances trained.
- *Different dataorder*: networks were trained with the exact same training data, however the order of the samples was varied for each model by choosing a different random seed before initialisation of the data loader; five instances trained.
- *Different architectures*: we trained a DenseNet-121 as a different architecture. Due to hardware constraints, we had to use a batch size of 64 for this condition; one instance trained.
- *Half data*: the network was trained on only half of the data but compared to the base condition with all data; one instance trained.
- *Combined condition*: for this condition, we combined multiple conditions. Here, we trained networks of the different architecture condition with training data in a different order, using SGD with Nesterov momentum, varying learning rates from 0.148 to 0.152, and different initialisations for each network; 5 instances trained.
- *Different data*: two networks were trained on the first and second half of the ImageNet training set respectively. Thus two different, *disjoint* training datasets were used—of course from the same distribution (ImageNet). For this condition we compared the networks to each other instead of comparing against the base condition.

## A.2    SOFTWARE, HARDWARE AND DATA

The networks were trained on GeForce RTX 2080 Ti GPUs with CUDA Version 11.1, CPU cores and 32 GB RAM shared between the cores. All code was written in PyTorch using Python 3 and the code to reproduce our findings is available in the supplementary material. For the RSA analysis, we used the thingsvision toolkit (Muttenthaler and Hebart, 2021). We used three data sets: ImageNet (Russakovsky et al., 2015), CIFAR-100 (Krizhevsky and Hinton, 2009) and the third dataset ("Gaussian noise") was generated by ourselves to investigate the effect of training on a dataset that does not contain any "natural image structure". It was generated by drawing pixel-wise uncorrelated Gaussian noise for each of the three RGB-channels. The dataset consisted of 100 classes with 20000 train and 50 test images per class. The $i$-th class has a mean of 128 and a standard deviation of $\sigma = i$, which is how classes can be identified by a model.

## A.3    CONTROL EXPERIMENTS WITH VGG AND DENSENET

To ensure that our findings generalize across different architectures and different datasets, we reran our main experiment with a number of variations:

---

[6]See `https://github.com/pytorch/examples/tree/master/imagenet`: batch size of 256, 90 epochs, the SGD optimizer and an initial learning rate of 0.1 that was divided by 10 every 30 epochs.

First, we tested different architectures; *ImageNet with Densenet-121 as base network*: Using a Denenet-121 as base network with a slightly altered training paradigm using only 30 instead of 90 epochs—to reduce the environmental impact of our study—and a batch size of 64 due to GPU RAM limitations. A ResNet-50 was used as comparison architectures.

*Imagenet with VGG-11 as base network*: For VGG-11 as the base network, we used a starting learning rate of 0.1 as according to the standard PyTorch implementation. Again, we only trained networks in this paradigm for 30 epochs and with learning rate steps every 10 epochs. Additionally, we used an AlexNet as different architecture.

Second, in addition to ImageNet and our Gaussian dataset we used another dataset, namely *CIFAR-100*: Again, we followed the standard ResNet-18 PyTorch implementation with the modification that we only used a total of 30 epochs to reduce the environmental impact of our study.

## A.4  Control experiment Representational Similarly Analysis

Additionally, to check whether our results are reproducible outside of a behavioural measure, we applied the tool representational similarly analysis (RSA). RSA is a method that originated in the brain sciences. It quantifies whether the inner representation—here the activation of kernels by single images—is similar across networks (Kriegeskorte et al., 2008; Mehrer et al., 2020). An RSA between two networks yields a correlation index between -1 and 1, indicating anti-correlation, no correlation (0) and perfect correlation respectively. It is important to note that the correlation values from RSA and $\kappa$ from error consistency are not comparable, although they have the same limits.

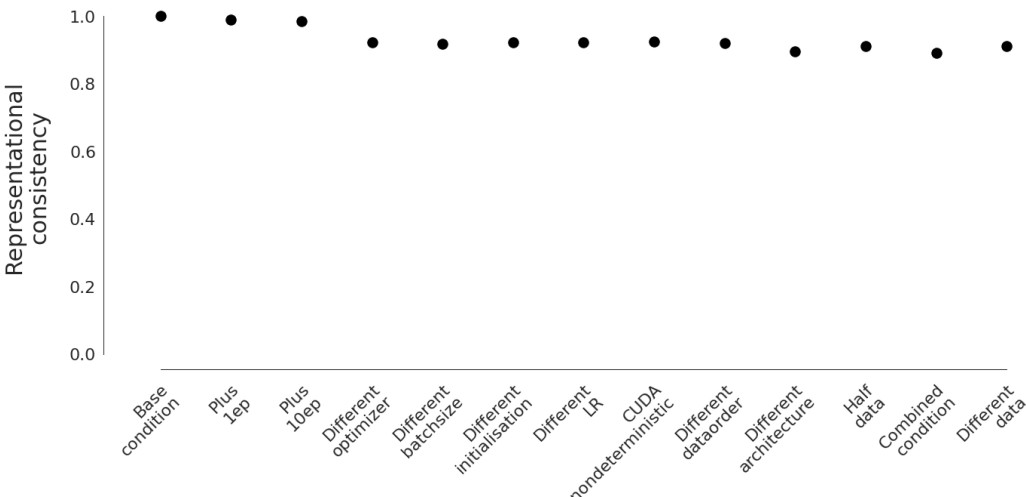

Figure 7: Correlations between the last fully connected layers of the different conditions and the base network on the ImageNet validation set after 90 epochs. For conditions in which multiple models were trained, only the first model was used.

## A.5 ANALYSING A SINGLE NETWORK

Figure 8 shows for our base network, whether ImageNet validation images are classified correctly (white) or incorrectly (blue) across epochs. There are three take-aways from this visualization. (1.), one immediately notices the influence of the standard learning rate steps after 30 and 60 epochs. However, after this step, some images (bottom) are also "forgotten" (classified correctly before step but incorrectly afterwards), which contrasts with the usual expectation that a model gradually improves over time. (2.), some images are learned immediately during the very first epoch and never forgotten later (top right region), while some are never learned at all. We will later see that this is not an effect of label errors, see Figure 2a. (3.), while accuracy usually only improves minimally from one epoch to the next (e.g. 0.04% from epoch 89 to epoch 90, or 14 additionally correctly classified images out of 50,000), on average 12.37% of the models' image classification decisions swap every epoch, corresponding to 6,184 images! (See Figure 17 in the Appendix for a plot which shows the number of swapped labels from epoch to epoch). The key takeaways from Figure 8 are already known from previous works investigating model errors over training time (Toneva et al., 2018; Mangalam and Prabhu, 2019; Kalimeris et al., 2019)—we do not intend to claim any conceptual novelty in this regard, Figure 8 simply intends to visualise these intriguing patterns clearly.

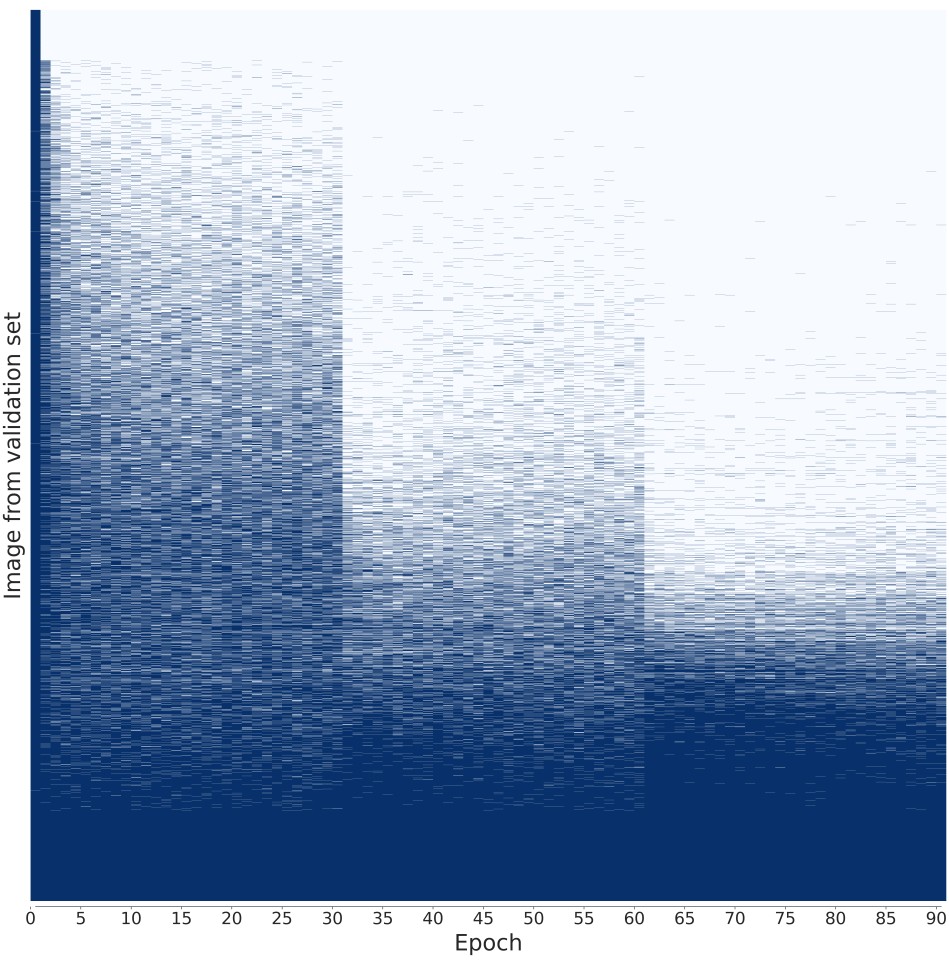

Figure 8: Decisions on all 50K ImageNet validation images of the *single* base network over the epochs. Blue indicates that the respective item was falsely classified during the specific epoch, while white indicates that it was correctly classified. The items from the ImageNet validation set are ordered according to the mean accuracy the base network achieved on them over the course of the 90 epochs. Therefore, items which were classified correctly from epoch 1 are on top and items which were classified incorrectly from epoch 1 are on the bottom.

## A.6 DDD IN CIFAR AND THE GAUSSIAN DATASET

Is dichotomous data difficulty (DDD) only a problem for ImageNet? We here show that this is not the case on two different datasets. CIFAR-100 (Krizhevsky and Hinton, 2009) and the third dataset ("Gaussian noise") was generated by ourselves to investigate the effect of training on a dataset that does not contain any "natural image structure". It was generated by drawing pixel-wise uncorrelated Gaussian noise for each of the three RGB-channels. The dataset consisted of 100 classes with 20000 train and 50 test images per class. The $i$-th class has a mean of 128 and a standard deviation of $\sigma = i$, which is how classes can be identified by a model[7].

As a first indication, for both of these datasets we find similarly high error consistencies between different models, just like we found for ImageNet (see Figure 21). Furthermore, training models on CIFAR-100 and the Gaussian data set leads to a very similar result pattern as for natural data sets like ImageNet and CIFAR-100 (shown in panel (b) and (c) of Figure 22). This is a strong indication —together with the imbalanced class accuracies in Figure 20— that highly consistent model errors are a result of DDD and not an artefact of natural images.

## A.7 KL DIVERGENCE

We constructed a third dataset ("Gaussian noise"). It was generated by drawing pixel-wise uncorrelated Gaussian noise for each of the three RGB-channels. The dataset consisted of 100 classes with 20000 train and 500 test images per class. The $i$-th class has a mean of 128 and a standard deviation of $\sigma = i$, which is how classes can be identified by a ML model. With this procedure, the KL-Divergence

$$KL(Class_i, Class_{i+1}) = log(\frac{\sigma_{i+1}}{\sigma_i}) + \frac{\sigma_i^2}{2 \cdot \sigma_{i+1}^2} - \frac{1}{2} \qquad (1)$$

between class $i$ and $i - 1$ is decreasing, see Figure 9.

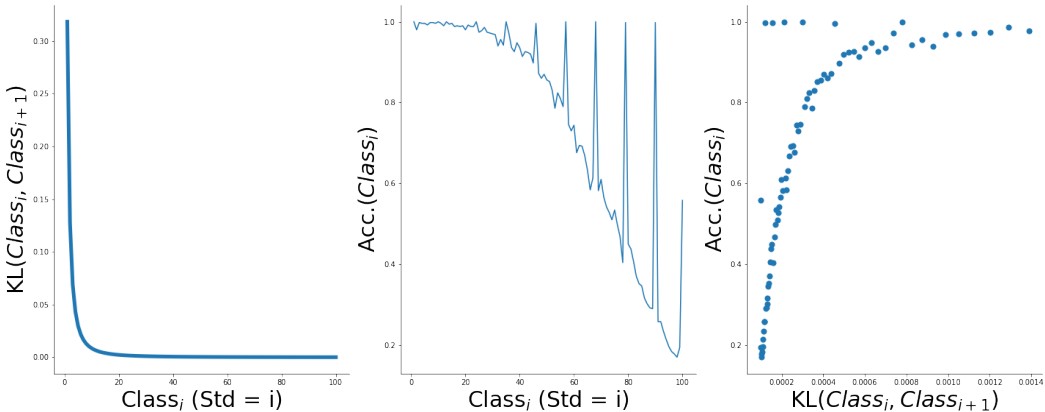

Figure 9: Kl-Divergence vs. accuracy for the Gaussian dataset. (Left) KL-divergence between $Class_i$ and $Class_{i+1}$. (Centre) Acc. of $Class_i$. (Right) Scatterplot between KL-divergence and accuracy. For the last plot we skip the first 20 classes (with accuracy close to 1) for better visibility.

## A.8 CLASS ACCURACIES

Figure 20 clearly shows that for all datasets, some classes are *very* easy to classify (e.g. up to 100% top-1 accuracy on ImageNet), while other classes are *very* difficult (e.g. down to 10% top-1 accuracy on ImageNet, for a list of top-10 easiest and hardest classes see Table 1). This means that there are both easy and hard images as well as easy and hard classes.

---

[7]We constructed the dataset with a decreasing KL-divergence between classes. Thus some classes are easier than others. In fact, we show (Figure 9) that the KL divergence is a very good predictor for class accuracies.

| Highest accuracy | Lowest accuracy |
|---|---|
| 'earthstar' | 'screen, CRT screen' |
| 'yellow lady's slipper, yellow lady-slipper, Cypripedium calceolus, Cypripedium parviflorum' | 'velvet' |
| 'proboscis monkey, Nasalis larvatus' | 'sunglass' |
| 'Leonberg' | 'ladle' |
| 'freight car' | 'tiger cat' |
| 'echidna, spiny anteater, anteater' | 'notebook, notebook computer' |
| 'African hunting dog, hyena dog, Cape hunting dog, Lycaon pictus' | 'hook, claw' |
| 'limpkin, Aramus pictus' | 'cleaver, meat cleaver, chopper' |
| 'hamster' | 'letter opener, paper knife, paperknife' |
| 'three-toed sloth, ai, Bradypus tridactylus' | 'spatula' |

Table 1: Table displaying the ten classes, for which the base network achieved with highest and lowest accuracies respectively. Items are in a descending order, so that 'earthstar' has the highest accuracy and 'screen, CRT screen' has the lowest accuracy.

## A.9    MODELING IMAGE DIFFICULTY

We modelled the image difficulty in Figure 2 as a delta peak, all images have the same difficulty. One of our reviewers suggested to model the image difficulty with an exponential decay instead—this means that we assumed that there are many simple images, then less and less more difficult images.

We therefore extend our previous approach by simulating an exponential and a sigmoid function. For each of the two functions, we binned the "difficulty" in bins of size 0.01 in the range of 0.00 to 1.00. We then calculated how many images are to be expected in each of the difficulty bins given the underlying function. For each of the difficulty bins, we then sampled from a binomial distribution. The mean difficulty of the images corresponds to the observed mean difficulty (p=0.689) for ResNet-18 variants. Both functions still yield very different histograms compared to the observed histogram of image difficulties. In order to better reproduce the observed histogram, a U-shaped function (bimodal distribution) is required.

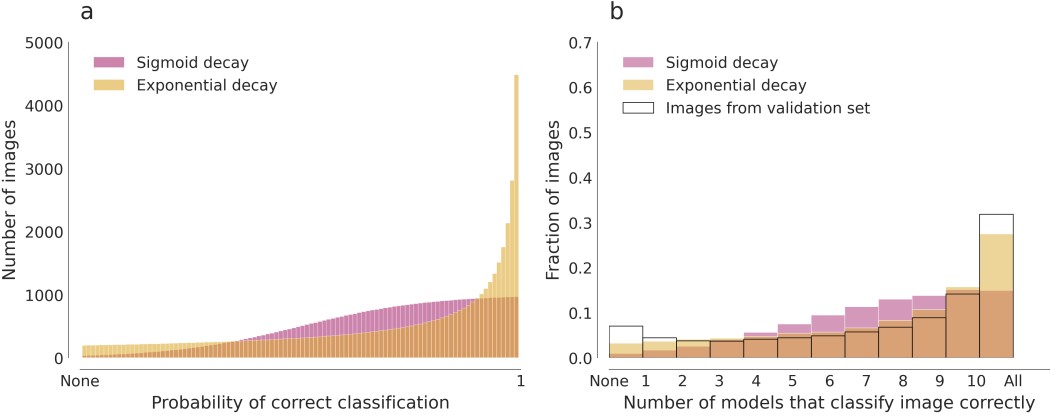

Figure 10: DDD is neither explained by an exponential/sigmoidal decay in image difficulty, nor by uniform example difficulty. (a) Binned functions modelling (exponentially) decaying (orange) and sigmoidal (purple) image difficulty. (b) Observed histogram (blue) with histograms obtained by sampling with a binomial observer given the exponential (orange) and sigmoid (purple) functions.

### A.10 METHODS OF THE HUMAN EXPERIMENTS

**Experiment 1: Can humans distinguish trivials and impossibles?**
In order to test whether humans can infer which images are easy and hard for CNNs, we conducted a psychophysical two-alternative forced choice experiment (Wichmann and Jäkel, 2018). In the experiment, observers were instructed to indicate by button press which image of an image pair they believe to be more difficult for a network to classify correctly—Is the right or the left image easier to classify for a CNN? An example trial can be found in the appendix, see Figure 23. Images were chosen from the ImageNet validation set such that the image pairs consisted of one image which all networks with different inductive bias classified correctly and another image which all networks misclassified (see also Figure 18). Stimuli were non-normalized images of size $224 \times 224$ px. Observers performed 149 self-paced trials. Overall, nine observers (mean age = 34.6 yrs, 2 female, 7 male) participated. Two observers were entirely naïve to CNN research, a further four were naïve to the purpose of the experiments, but knew about CNNs. Subjects received monetary compensation of 10 € per hour. The total duration of the experiment was 30 minutes per observer.

**Experiment 2: What makes trivial, inbetween and impossible images different?**
Thanks to the suggestion of our reviewers, we designed a follow-up experiment to better understand the differences between "trivials", "impossibles" and "in-betweens". We randomly chose 100 "trivial" (all networks give the correct response), 100 "impossible" (no network gives a correct response) and 100 "in-between" images. Each observer had to answer the following questions for each image:

- Q1: How many objects belonging to different categories are in the image (e.g. three dogs are still one category: dog. But two dogs and one cat are two categories: dog and cat)?

- Q2: Is there a main category in the image? (No, maybe, Yes)

- Q3: Is the presentation of the objects unusual in any manner (e.g. orientation, location, size, viewpoint)? (No, slightly, very)

Stimuli were non-normalised images of size $224 \times 224$ px on a white background with the trial number on the left, see example image in Figure 24. All observer rated the same 300 images.

Overall, 8 observers (mean age = 36.0 yrs, 1 female, 7 male) participated. Two observers were entirely naïve to CNN research, a further three were naïve to the purpose of the experiments but have experience in working with CNNs and three observers were non-naïve. The total duration of the experiment was 90 minutes per observer. Subjects received compensation worth 15€.

## A.11 CONSISTENCY IN THE SECOND HUMAN PSYCHOPHYSICAL EXPERIMENT

Due to a suggestion of our reviewers, we checked the consistencies between our observers in this follow-up experiment . For question one (number of categories), the average pairwise correlation was 0.62 [min = 0.42 , max = 0.76]. For question number two (is there a main category) and three (presentation oddness) we calculated Krippendorff's alpha (0 indicates no agreement between raters, 1 implies perfect agreement [8]) which is suited for ordinal data. For question two, we also removed subject two from the analysis, since they reported that they did not use *object* categories but instead semantic categories like playing music or partying in the debriefing. Here we obtained on average of $\alpha_2 = 0.50[min = 0.38, max = 0.63]$ for question 2 and $\alpha_3 = 0.33[min = 0.10, max = 0.48]$ for question 3. Furthermore, we made sure that these numbers are not affected by the pre-knowledge of the observers. Thus there was reasonably high agreement between the observers despite the rather open (or vague) nature of the task and instructions.

---

[8]Krippendorff offers in his book (Krippendorff, 2004, p. 241) a lower bound only for very high agreement.

A.12 SUPERCLASS ANALYSIS

We analysed the proportions of images from the three image categories ("trivials", "in-betweens" and "impossibles") belonging to each of the ImageNet superclasses. The superclasses result from the WordNet hierarchy and are sometimes also referred to as subtrees Deng et al. (2009). We mapped the unique image identifiers from the ImageNet validation images to their respective superclasses using a file from the git repository of Tsipras et al. (2020).

This analysis is visualised in Figure 11. Here we show the distribution of image subsets ("trivial", "in-between", "impossible") within each superclass. Since we have less impossibles than trivial, they are overall more rare. The two most extremely unbalanced superclasses are "Implements, containers, misc. objects" and "Birds". For "Implements [...]", only 20% of images are in the "trivial" subset and 10% are in the "impossible" subset. In contrast, the "Birds" superclass consists of 60% "trivials" and <5% "impossibles". We therefore find that the image subsets are not equally distributed for the superclasses. Furthermore, we show that the superclasses are not equally distributed over the image subsets, see Figure 12. Both Figures point towards the same conclusion: there are easier and harder superclasses.

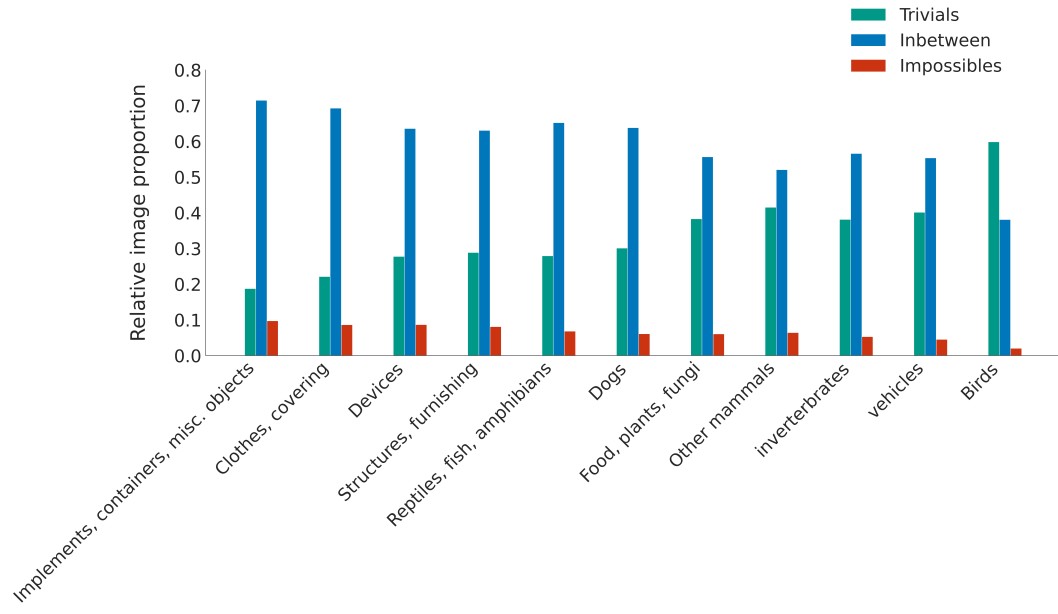

Figure 11: Barplot showing the proportions of items from each of the three image subsets ("trivial", "in-between", "impossible") belonging to each superclass. Here, the values are normalized so that the proportions of items in each superclass sum to 1. The superclasses are ordered according to the proportion of impossibles to trivials.

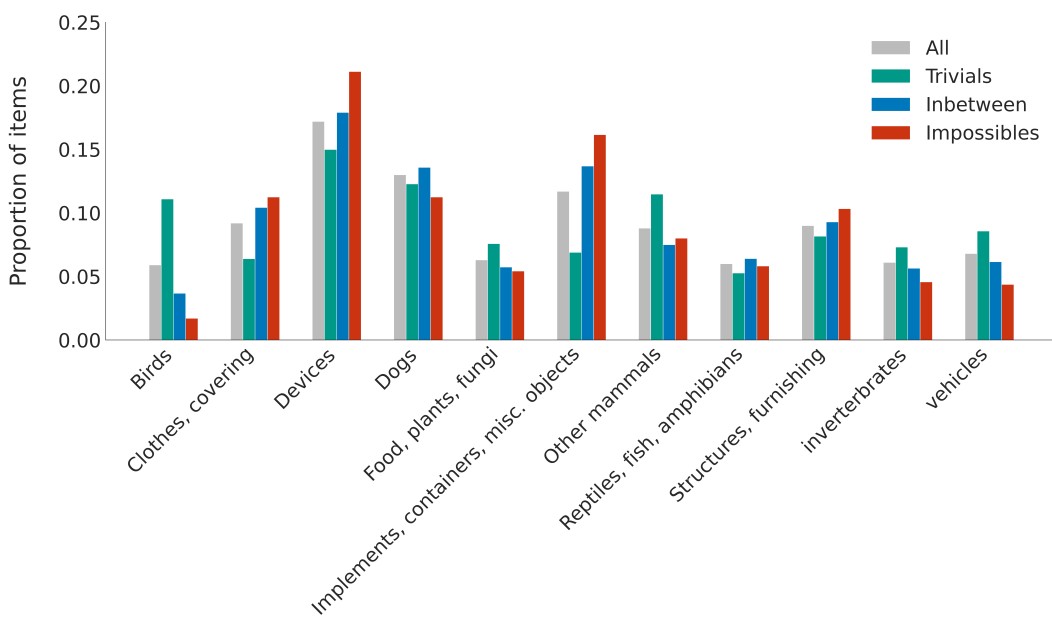

Figure 12: Proportion of items from each of the three image categories ("Trivials", "Inbetweens" and "Impossibles") belonging to each superclass. The values are normalized so that the proportions of each subset sum to 1.

### A.13 LABEL AMBIGUITY AS CAUSE FOR THE "IN-BETWEEN"

One of our reviewers suggested that label ambiguity might be a major cause for the emergence of "in-between" images.

We agree that there is a possibility that disagreement between human annotators could reduce the accuracy for the "in-between" images. Label ambiguity is known to affect image datasets (Whitehill et al., 2009; Peterson et al., 2019; Gordon et al., 2021)—although interestingly the ImageNet creators tried to mitigate this, see Russakovsky et al. (2015, p.7 onward). We agree that label ambiguity can affect model accuracies. Thus we decided to investigate this hypothesis (label ambiguity as a cause for disagreement) using two different, independent datasets to analyse label ambiguity in ImageNet.

First we revisit the dataset of Northcutt et al. (2021b). The authors automatically detected label errors in the ImageNet validation set and used Amazon Mechanical Turk to manually check every possibly falsely labelled image with 5 human raters. If label ambiguity is a cause of the "in-between" we expect that the 5 human raters do not a agree on the "in-betweens". Thus we combine combine Northcutt's data and with our previous analysis. Overall, for the MTurk analysis Northcutt proposes label errors on 5440 images in the ImageNet validation set. From these 5440 potential label errors 2643 are in the "in-between" class. Out of the potential 2643 images, on 1945 at least one rater was not in agreement with the others. However please note that this high rate is expected, since the 2643 images are those already identified as possibly having ambiguous labels by the automatic approach. We have to compare this number to the total number of images in the "in-between" subset, which is 21248 images. Hence, only 9% (=1945/21448) of "in-between" images suffer from label ambiguity, compared to an overall rate of 8,8%(4424/50000) label ambiguity for the entire dataset.

This provides evidence that label (dis-)agreement may not be a main confounder in our experiment, but of course an automated approach might miss certain images. We therefore make use of another dataset: that of Geirhos et al. (2021) with humans (which does not rely on any automated assessment).

Geirhos et al. (2021) used four observers, which performed a classification task in a highly controlled psychophysical setting on a subset of the ImageNet validation set. In total the observers classified 607 colored images belonging to the "in-between" subset. Here the four observer agreed on 85% (513 out of 607) of the images. Thus, the disagreement between raters is fairly small.

Both papers used completely independent raters and different strategies (MTurk vs. highly controlled psychophysics). Still, analysing data from both approaches point towards the same result, providing evidence that only a minor fraction of "in-between" are affected by label ambiguity.

## A.14 FIGURES

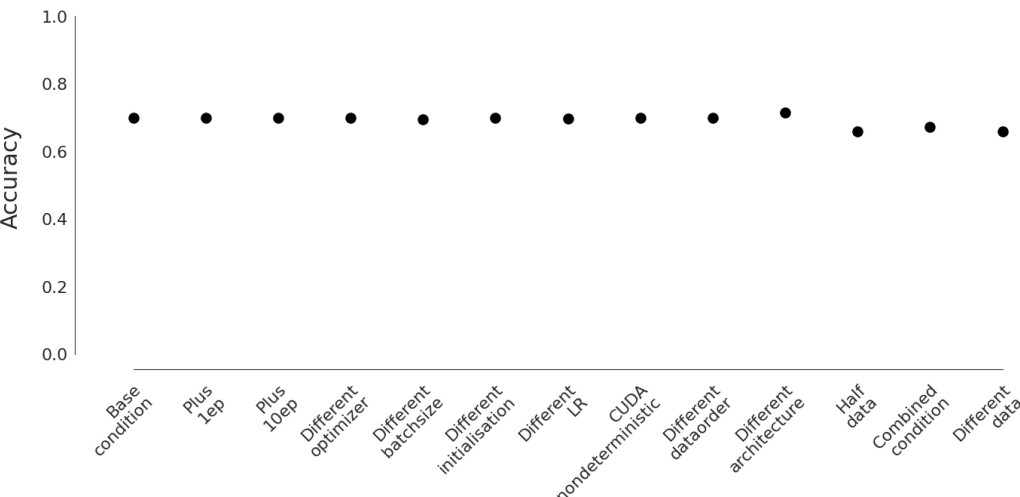

Figure 13: Accuracies of the different conditions and the base network on the ImageNet validation set after 90 epochs. The mean over all models of a condition is displayed here.

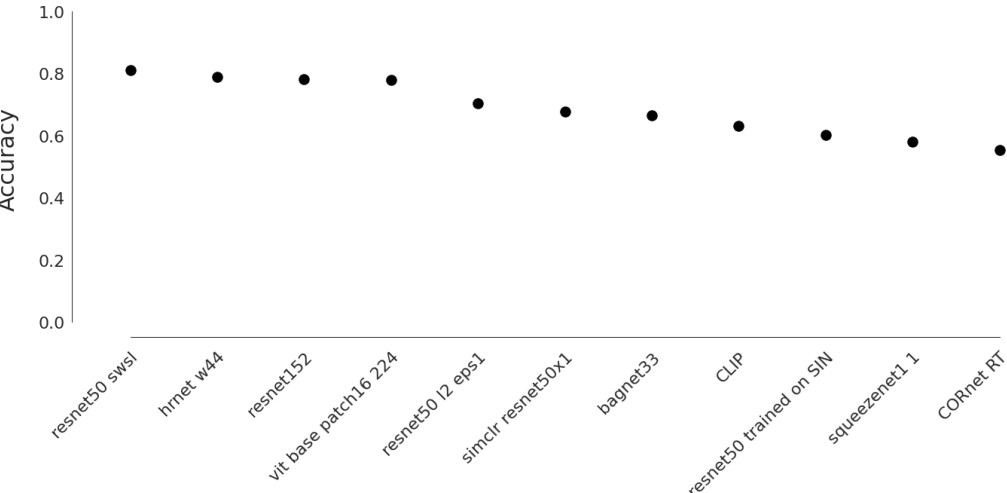

Figure 14: Accuracies for the SOTA models on the ImageNet validation set. Mean accuracy of all models is 0.689

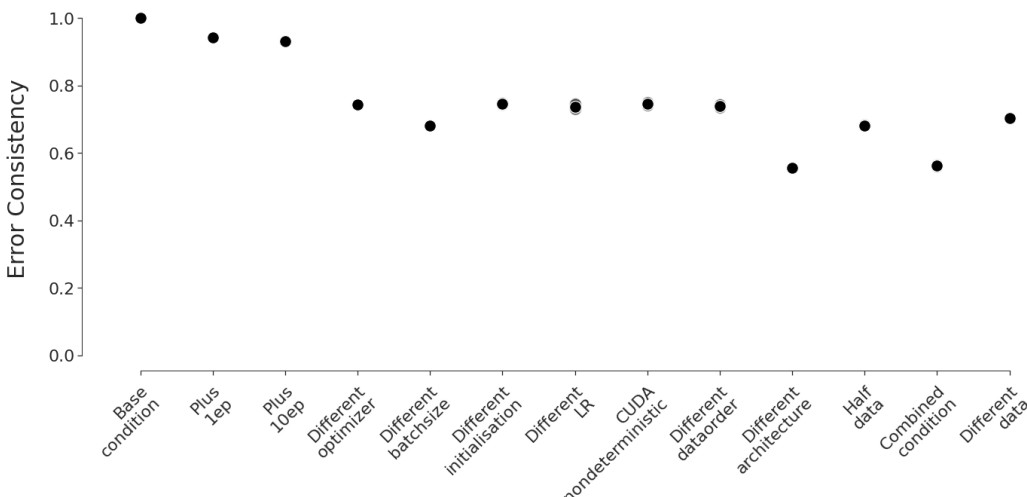

Figure 15: Error consistencies on the ImageNet validation set with VGG-11 as the base network. All variations performed are the same as outlined in the Methods section. In this case, the different architecture is an AlexNet. The conditions are ordered by the mean error consistency on the ImageNet validation set for ResNet-18 as the base network (see Figure 3). For conditions in which multiple models were trained, the model-wise error consistencies are plotted with a lower opacity compared to the mean over all models for the conditions.

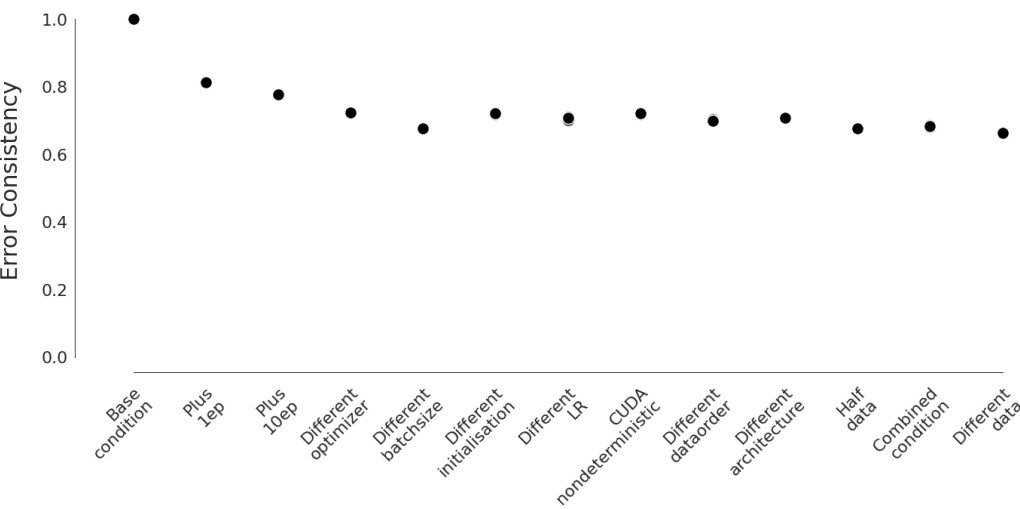

Figure 16: Error consistencies on the ImageNet validation set with DenseNet-121 as the base network. All variations performed are the same as outlined in the Methods section. In this case, the different architecture is a ResNet-50. The conditions are ordered by the mean error consistency on the ImageNet validation set for ResNet-18 as the base network (see Figure 3). For conditions in which multiple models were trained, the model-wise error consistencies are plotted with a lower opacity compared to the mean over all models for the conditions.

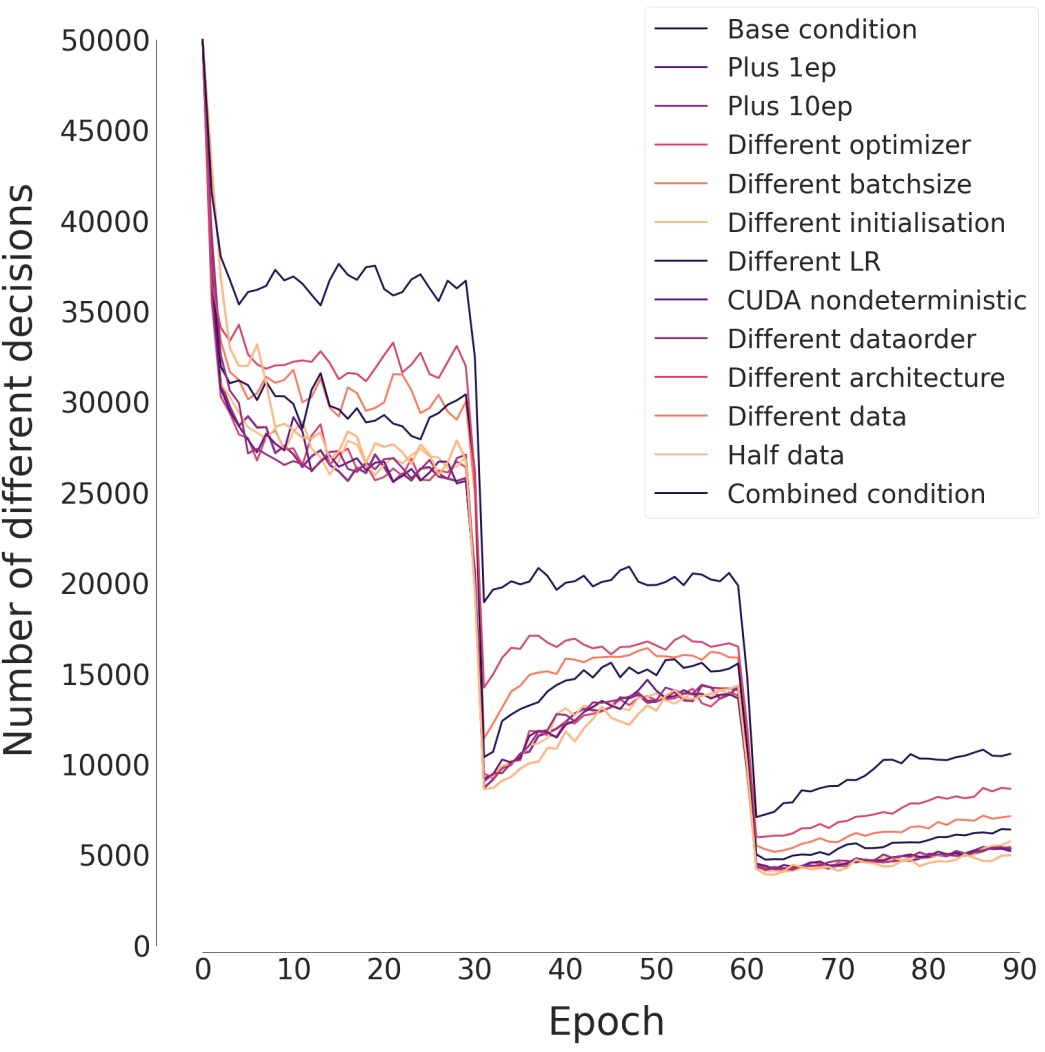

Figure 17: Lineplot showing the number of decisions that change from the current to the following epoch. For epoch 0, this means that the number of decisions that are different between epoch 0 and epoch 1 are shown. For conditions in which multiple model instances were trained, only the last instance is shown for the sake of simplicity.

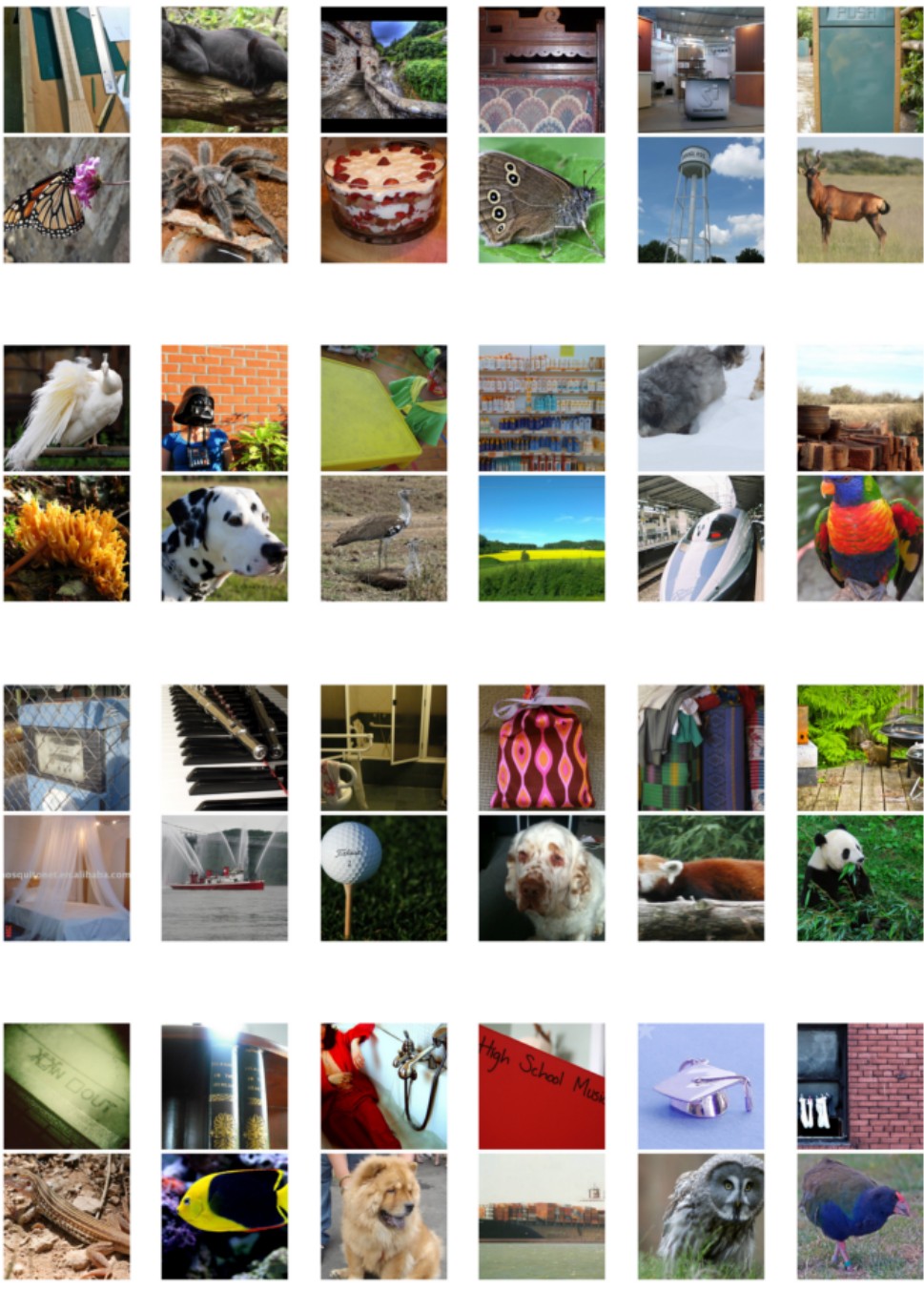

Figure 18: Pairs of impossible (top) and trivial images (bottom) from ImageNet.

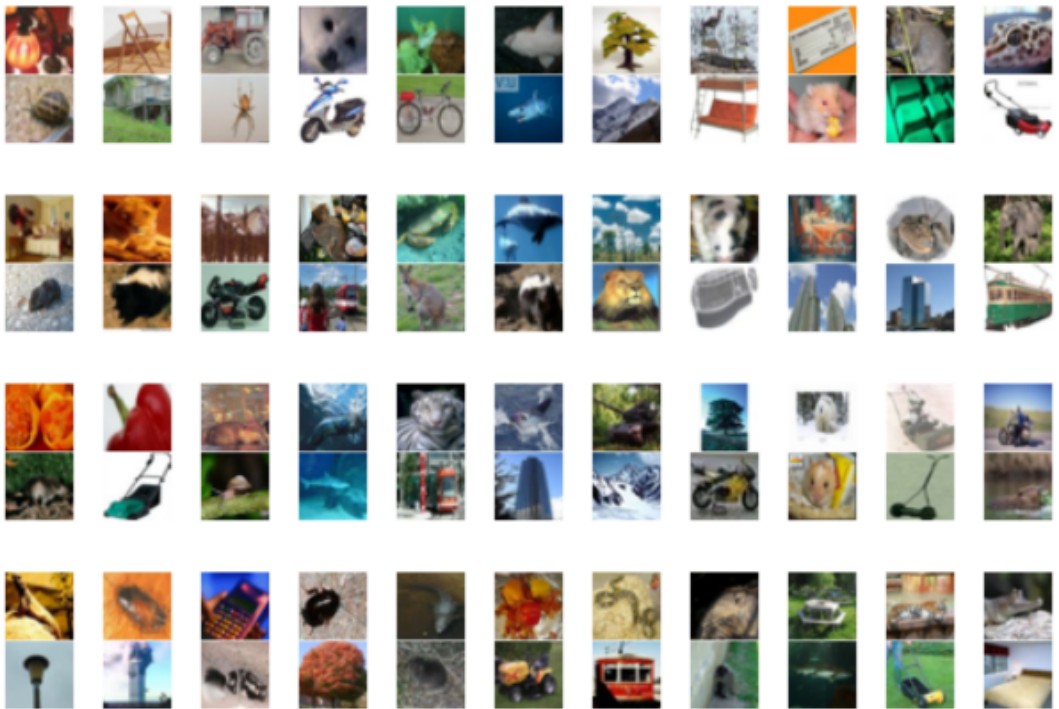

Figure 19: Pairs of impossible (top) and trivial images (bottom) from CIFAR-100.

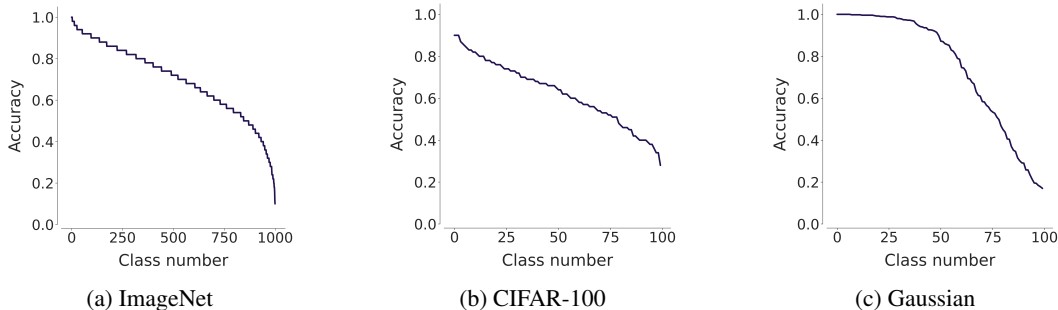

(a) ImageNet

(b) CIFAR-100

(c) Gaussian

Figure 20: Class-wise accuracy per dataset. Shown is the decreasing accuracy for all classes in the validation sets and for the fully trained base network.

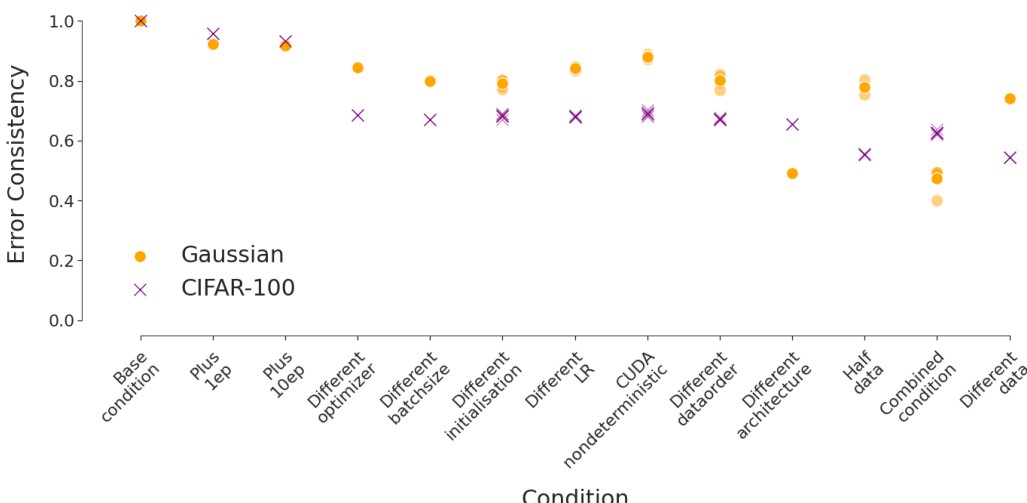

Figure 21: Error consistencies between the different conditions and the base network for the validation sets of CIFAR-100 and our Gaussian dataset. The conditions are ordered by the mean error consistency on the ImageNet validation set (see Figure 3). For conditions in which multiple models were trained, the model-wise error consistencies are plotted with a lower opacity compared to the mean over all models for the conditions.

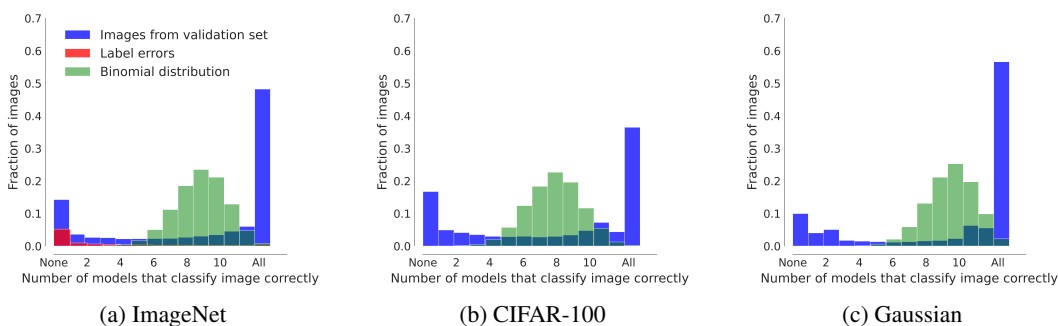

Figure 22: Histogram showing how many models correctly classify validation sets images in the last epoch. In blue, the densities of how many items were answered correctly are shown. "None" indicates that no models classified the items correctly (impossibles), while for "All" items were classified correctly by all models ("trivial images"). For the sake of simplicity, only the last model was used for conditions where multiple models were trained. In green, samples are drawn from a binomial distribution with $n$ equal to the number of models and $p$ equal to the mean accuracy over the models. Additionally for ImageNet, the distribution of 5000 label errors as identified by the cleanlab package are shown in red (Northcutt et al., 2021a).

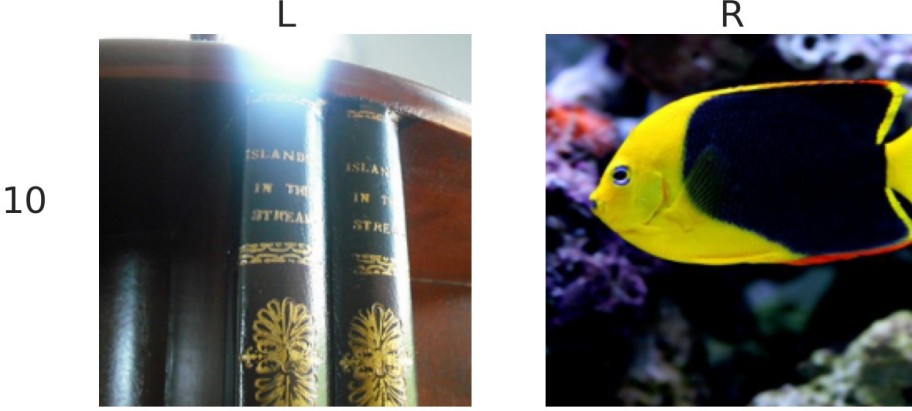

Figure 23: Example trial from the first psychophysical experiment. Observers were asked: "Is the right or the left image easier to classify for a neural network?". The number on the left indicates the trial number and the letters "R" and "L" above the images were entered into the answer sheet by the observers.

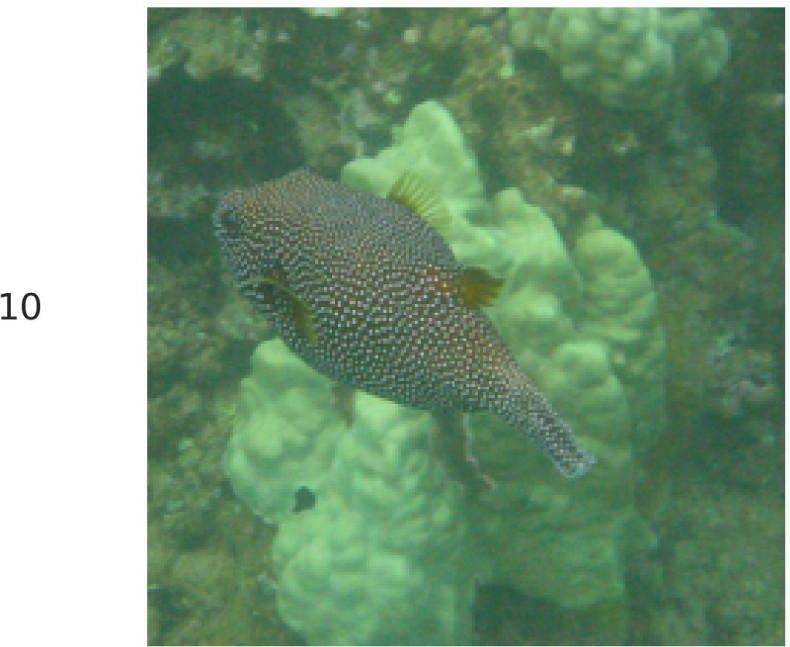

Figure 24: Example trial from the second psychophysical experiment. Observers were asked: "How many objects belonging to different categories are in the image (e.g. three dogs are still one category: dog. But two dogs and one cat are two categories: dog and cat)?", "Is there a main category in the image? (No, maybe, Yes)" and "Is the presentation of the objects unusual in any manner (e.g. orientation, location, size, viewpoint)? (No, slightly, very)". The number on the left indicates the trial number.

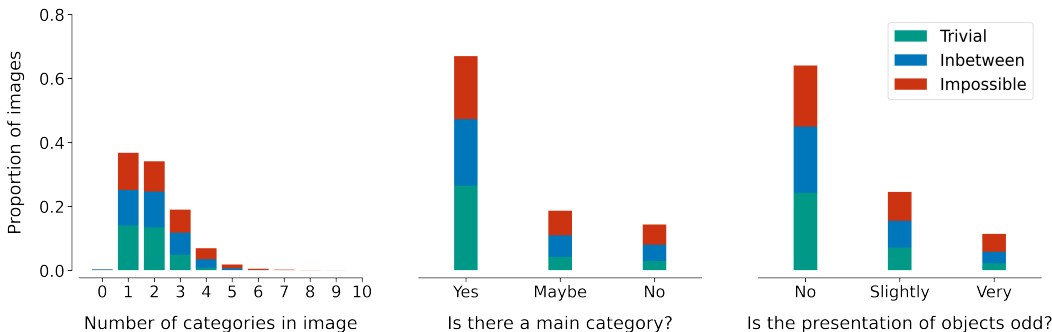

Figure 25: Barplot displaying the proportions of answers over all observers. We did not remove label errors for this plot. The bars are normalized so that the proportions of the different answers add up to 1 for each question.

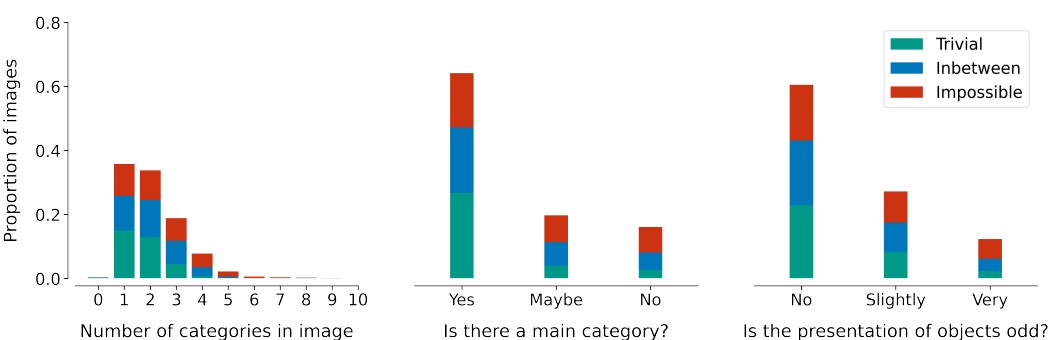

Figure 26: Barplot displaying the proportions of answers over all observers. For this plot, we removed images which were found to have label errors by Northcutt et al. (2021b) and balanced the image classes to have the same number of images. The bars are normalized so that the proportions of the different answers add up to 1 for each question.

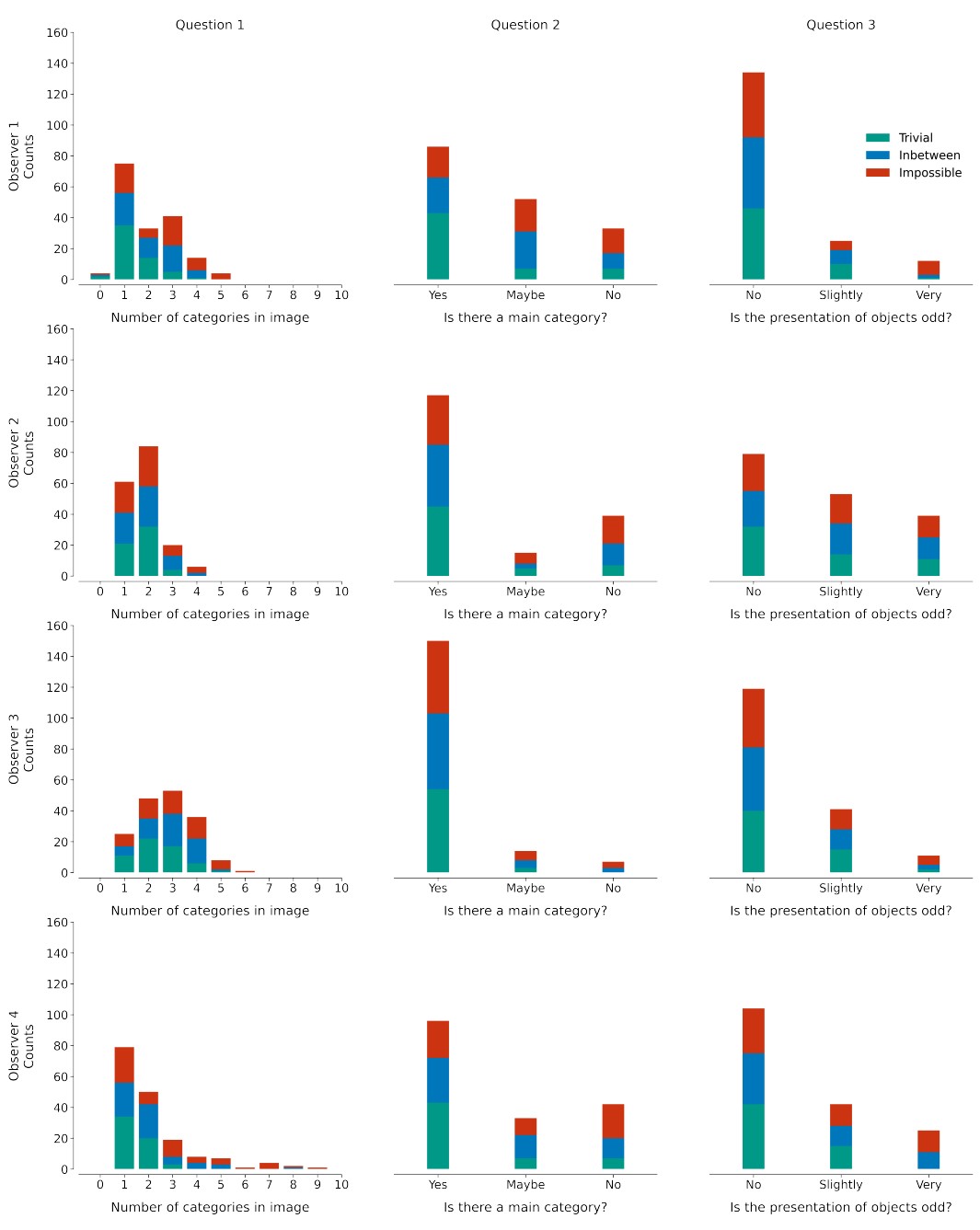

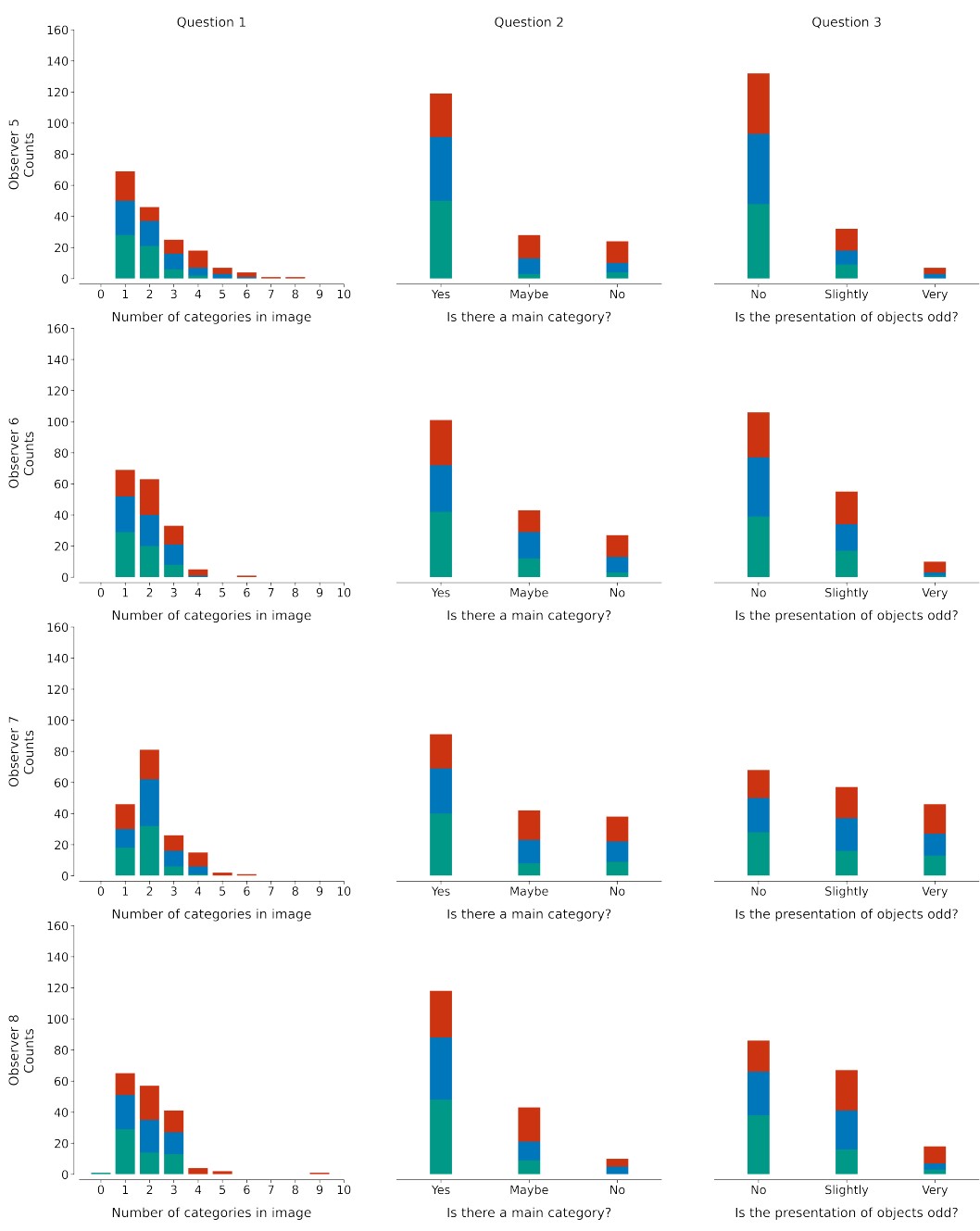

Figure 27: Barplots displaying the proportions of answers for each individual observer. We did not remove label errors for this plot. The bars are normalized so that the proportions of the different answers add up to 1 for each question.

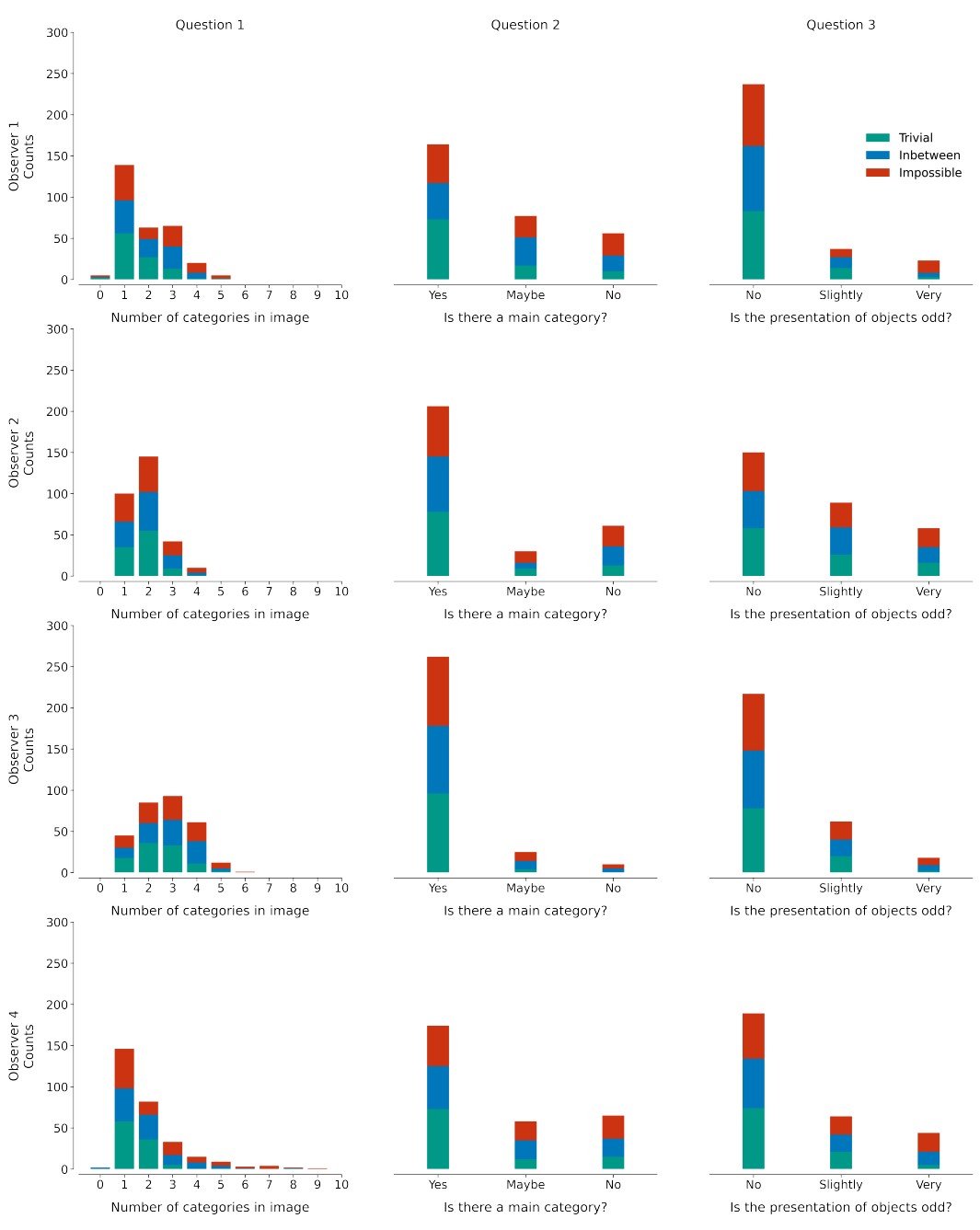

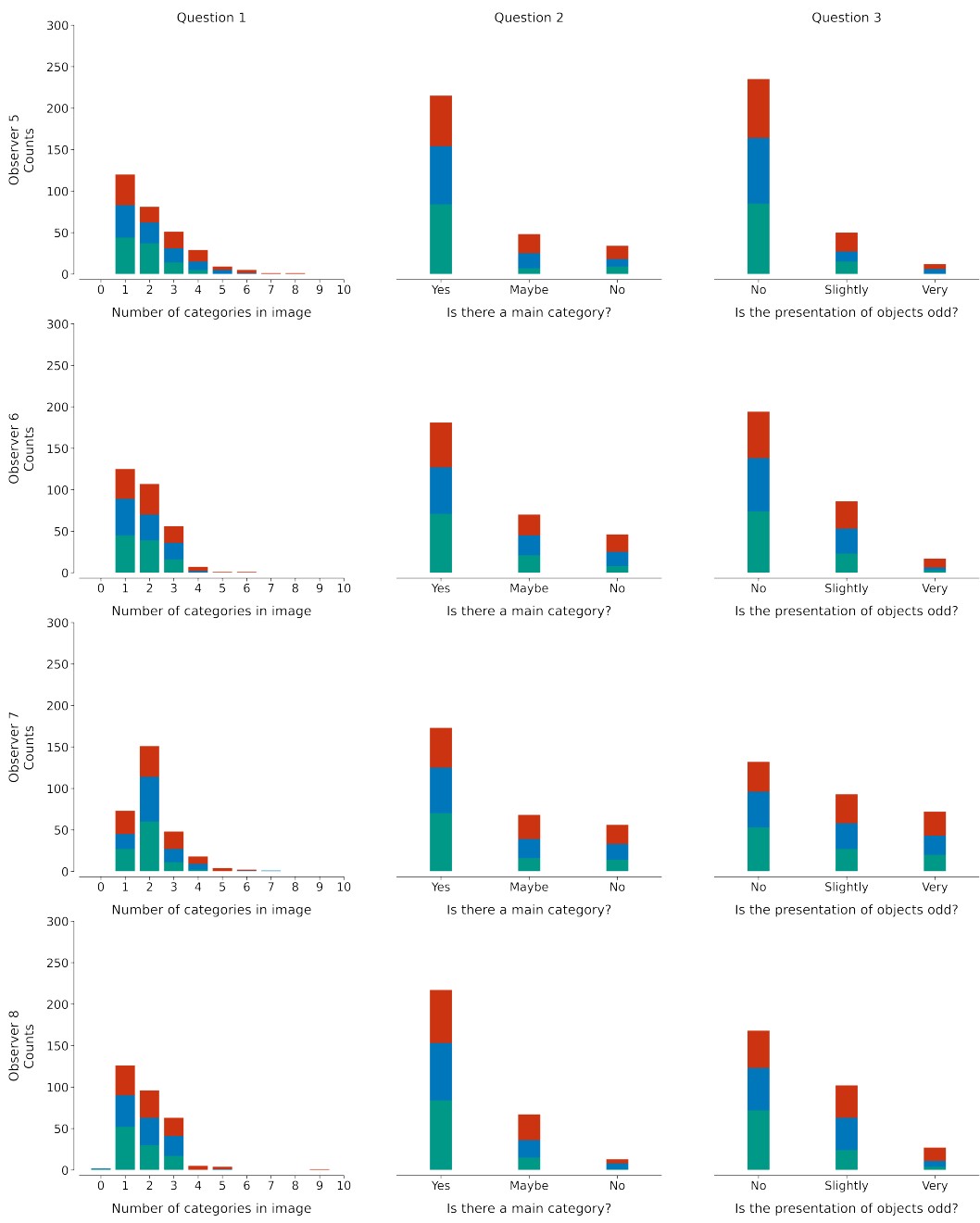

Figure 28: Barplots displaying the proportions of answers for each individual observer. We removed images which were found to have label errors by Northcutt et al. Northcutt et al. (2021b). The bars are normalized so that the proportions of the different answers add up to 1 for each question.

