# OpenReview forum: "Trivial or Impossible --- dichotomous data difficulty masks model differences (on ImageNet and beyond)"
_ICLR.cc/2022/Conference — ICLR 2022 Poster_

### Official Review · Reviewer_3xZg · 2021-10-29

**Correctness:** 2
**Technical Novelty And Significance:** 3
**Empirical Novelty And Significance:** 3
**Recommendation:** 6
**Confidence:** 3

**Main Review:**

Strengths:
1. The authors tackle a timely and important problem.
2. I found the paper to be relatively clearly written and concise.
3. The experiments that the authors conduct are fascinating and in my view point to many potential challenges in today's evaluation of image classification models which must be further explored.

Weaknesses:
Overall, I wish the authors had focused more on what seems to me as the most supported claim the authors can make given their experiments: around half of the images in existing test sets are not providing meaningful signal about the performance of today's models, and the consequence of this might be that existing evaluation approaches are hiding or at least downplaying the differences between models. All the authors show is that within this "difficult" set, models make different errors. Are these errors just randomly distributed between all the images? Are they clustered around certain classes? If we just focus on this "difficult" set, how do standard classification metrics differ between models?

To me, the claims the authors make about "difficulty" are far less supported and require significant additional work. As it stands, the wording used by the authors may be potentially misleading. How can we be confident that these images are "difficult"? Isn't it possible that these images may simply contain multiple possible plausible labels, and annotators disagreed? Existing work has shown relatively high disagreement rates [1], [2] in image classification datasets among annotators.

[1] Joshua C. Peterson*, Ruairidh M. Battleday*, Thomas L. Griffiths, & Olga Russakovsky (2019). Human uncertainty makes classification more robust. In Proceedings of the IEEE International Conference on Computer Vision.
[2] Mitchell L. Gordon, Kaitlyn Zhou, Kayur Patel, Tatsunori Hashimoto, and Michael S. Bernstein. 2021. The Disagreement Deconvolution: Bringing Machine Learning Performance Metrics In Line With Reality. Proceedings of the 2021 CHI Conference on Human Factors in Computing Systems.

The authors also need to detail the exact procedure used in their human study. Exactly what questions were asked, what did the interface look like, how many participants were there, how were they compensated, etc. Also, why did the authors not ask annotators whether an image would fall into the 3rd "difficult" category?

**Summary Of The Paper:**

The authors demonstrate that, irrespective of model architecture or hyperparameters, many modern image classification models are always correct on 46.0% “trivial” and 11.5% “impossible” images. The authors describe the remaining 42.5% of images as "difficult", and claim that by focusing on these difficult images, it is possible to see pronounced differences between models. They also find that humans can
predict which images are “trivial” and “impossible” for CNNs at 81.4% accuracy.

**Summary Of The Review:**

The authors tackle an important topic and perform fascinating experiments, but there is insufficient evidence for the claims they make and, in my view, the claims they could better make were not deeply explored.

---

> ### Author Response · Authors · 2021-11-15
> **Author response to reviewer 3xZg**
>
> Dear reviewer 3xZg,\
> we are happy to hear that you find our paper to "tackle a timely and important problem" and to be "clearly written and concise". We are especially pleased to read that our experiments are "fascinating" and "point to many potential challenges in today's evaluation". Thank you very much for your constructive review!
>
> *Further analysis of the three classes (Are errors just randomly distributed between all the images? Are they clustered around certain classes? How can we be confident that these images are "difficult"? Isn't it possible that these images may simply contain multiple possible plausible labels, and annotators disagreed?)* \
> This is an excellent suggestion which is shared with the other reviewers. We address this question with the follow-up psychophysical experiment outlined above. We will get back to you with the results within a few days, after data collection and analyses are complete.
>
> *Exactly what questions were asked, what did the interface look like, how many participants were there, how were they compensated, etc. Also, why did the authors not ask annotators whether an image would fall into the 3rd "difficult" category*?\
> We highly appreciate your concerns about experimental reproducibility---this is also a very important aspect for us. In the previous version we already included information about the number of participants, compensation, etc. in the methods section. We now added the exact question that was asked ("Is the right or the left image easier to classify for a neural network?") as well as an example trial in the appendix (see Figure 20). Our research paradigm is a classical two alternative forced choice question (2-AFC). We used a 2-AFC paradigm since it has a higher sensitivity compared to a labelling task (see e.g. the cited paper of Wichmann & Jäkel, 2018), which is why we only use the two stimuli alternatives.

---

> > ### Comment · Reviewer_3xZg · 2021-11-18
> > **Response**
> >
> > My thanks to the authors for their response.
> >
> > You've addressed my reproducibility concerns, thanks!
> >
> > I think that your proposed experiment makes progress towards but isn't quite getting at the point I was trying to make. It's possible that multiple different annotators each might think there is only one class present in an image, but that they select something different for that one class. I'm not sure your experiment can directly tell us how often this is occurring.

---

> > > ### Author Response · Authors · 2021-11-22
> > > **Author response to reviewer 3xZg**
> > >
> > > Dear 3xZg,
> > >
> > > First of all, please find the result of our experiment as general comment above (https://openreview.net/forum?id=C_vsGwEIjAr&noteId=QTo4J6xgBVk).
> > >
> > > Thank you very much for your response. To clarify, you wrote *The authors describe the remaining 42.5% of images as "difficult"*. We thus assumed the images you refer to as "difficult" are those images we call "in-between". Furthermore, you wrote that *Isn't it possible that the difficult images ["in-betweens"] may simply contain multiple possible plausible labels, and annotators disagreed*?
> > >
> > > We agree that there is a possibility that disagreement between human annotators could reduce the accuracy for the "in-between" images. Label ambiguity is known to affect image datasets (Whitehill et al. 2009, Joshua et al. 2019, Gordon et. al. 2021)---although interestingly the ImageNet creators tried to mitigate this, see Russakovsky et al. (2015, p.7 onwards). We agree that label ambiguity can affect model accuracies. Thus we decided to investigate this hypothesis (label ambiguity as a cause for disagreement) using two different, independent datasets to analyse label ambiguity in ImageNet.
> > >
> > > First we revisit the dataset of Northcutt et al. (2021). The authors automatically detected label errors in the ImageNet validation set and used Amazon Mechanical Turk to manually check every possibly falsely labelled image with 5 human raters. If label ambiguity is a cause of the "in-betweens" we expect that the 5 human raters do not a agree on the "in-betweens". Thus we combine combine Northcutt's data and with our previous analysis.
> > >
> > > Overall, for the MTurk analysis Northcutt proposes label errors on 5440 images in the ImageNet validation set. From these 5440 potential label errors 2643 are in the "in-between"class. Out of the potential 2643 images, on 1945 at least one rater was not in agreement with the others. However please note that this high rate is expected, since the 2643 images are those already identified as possibly having ambiguous labels by the automatic approach. We have to compare this number to the total number of images in the "in-between" subset, which is 21248 images. Hence, only 9% (=1945/21448) of "in-between" images suffer from label ambiguity, compared to an overall rate of 8,8%(4424/50000) label ambiguity for the entire dataset.
> > >
> > > This provides evidence that label (dis-)agreement may not be a main confounder in our experiment, but of course an automated approach might miss certain images. We therefore make use of another dataset: that of Geirhos (2021) with humans (which does not rely on any automated assessment). Geirhos et al. (2021) used four observers, which performed a classification task in a highly controlled psychophysical setting on a subset of the ImageNet validation set. In total the observers classified 607 colored images belonging to the "in-between" subset. Here the four observer agreed on 85% (513 out of 607) of the images. Thus, the disagreement between raters is fairly small.
> > >
> > > Both papers used completely independent raters and different strategies (MTurk vs. highly controlled psychophysics). Still, analysing data from both approaches point towards the same result, providing evidence that only a minor fraction of "in-betweens" are affected by label ambiguity.
> > >
> > > Thanks again for raising this important question. Please let us know whether this analysis (which we now include in the Appendix, section A. 11) has addressed your concerns!

---

> > > > ### Comment · Reviewer_3xZg · 2021-11-24
> > > > **Response**
> > > >
> > > > My thanks to the authors for their thorough response! Given the data they've provided, I agree with their assessment that annotator disagreement is likely not a primary factor in explaining difficult images. I've decided to raise my score to a 6. My remaining concerns were well articulated by reviewer MrqK.

---

> > > > > ### Author Response · Authors · 2021-12-01
> > > > > **Thank you for your feedback!**
> > > > >
> > > > > Dear 3xZg,
> > > > >
> > > > > thank you very much for increasing your score. The discussion regarding annotator agreement clearly made our paper stronger. We will make sure to address the concerns by MrqK in the final version.

---

### Official Review · Reviewer_rhYo · 2021-10-31

**Correctness:** 4
**Technical Novelty And Significance:** 4
**Empirical Novelty And Significance:** 3
**Recommendation:** 6
**Confidence:** 5

**Main Review:**

Strengths:
1.	Thorough and solid empirical studies on the decision consistency between CNNs trained on the same dataset.
2.	Clear analysis showing the bimodal distribution of consistency across images. One large part is almost trivial for all CNNs and the other small part is almost impossible. The middle part is what differentiates the CNNs.
3.	The authors further show deeper analysis on the reason of this consistency through training models without the trivial and the impossible parts and showing that these models are much more different.
4.	The paper also shows that this consistency pattern happens for different datasets, including ImageNet, Cifar, and self-constructed Gaussian vectors.

Weaknesses:
1.	It seems to me that a very straightforward hypothesis about these two parts would be that the trivial part is what’s very simple, either highly consistent to what’s in the training set, or the images with very typical object pose in the center of the images; and for the impossible part, it might be the images with ambiguous labels, atypical object pose or position. I think the human test results would support this hypothesis, but I wonder whether the authors could provide more evidence to either prove or disprove this hypothesis.
2.	The figure 6 is very confusing to me. The caption says that the right part is original ImageNet test set, but the texts on the image actually say it’s the left part. If the texts on the image are right, then the right panel is the consistency on the validation images between the two parts. If I understand the experiments correctly, these results are for models trained on ImageNet training set without the trivial or the impossible part and then tested on ImageNet validation set without the two parts. Although it’s good to see the lower consistency, it should be compared to the consistency between models trained on the whole ImageNet training set and tested on ImageNet validation set without the two parts, which I cannot find. Is the consistency lower because of the changed training process or the changed validation set?
3.	It is also unclear how surprising we should be towards the consistency distribution, is this a result of an exponential distribution of the general “identification” difficulty (most images are simple, then less and less are more difficult)?


**Summary Of The Paper:**

This paper shows the high consistency between the decisions of CNNs trained on the same dataset, regardless of the algorithms, architectures, hyperparameters in training, and optimizers. The authors further show that the validation set of ImageNet contains a large part that’s “trivial” for all CNNs and a small part that’s almost “impossible”. Removing these parts in the training set indeed makes models more different. Finally, the authors state that humans can easily tell what images are “trivial” and what images are “impossible”.

**Summary Of The Review:**

The authors show the high consistency between the decisions of CNNs trained on the same dataset. It is also shown that the validation images can be split into three parts: the trivial part, the impossible part, and the part that’s between them. However, the current work lacks the test of a naïve hypothesis about these two parts: the trivial part is the easy images, and the impossible part is the difficult images. Some results are also not described clearly. I can only recommend for acceptance with confidence marginally above the threshold, but fixing these points would increase my confidence.

---

> ### Author Response · Authors · 2021-11-15
> **Author response to reviewer rhYo**
>
> Dear reviewer rhYo,
>
> we would like to thank you for your assessment of our work as a "thorough and solid empirical study". Especially your comment regarding Figure 6 clearly improved our paper. Thank you very much! In the following we will discuss the concerns you raised:
>
>
> *What leads to images being impossible, trivial or in-between (e.g. ambiguous labels, atypical object pose or position...)*?
> This is an excellent suggestion which is shared with the other reviewers. We address this question with a follow-up psychophysical experiment outlined above. We will get back to you with the results within a few days, after data collection and analyses are complete.
>
>
> *The figure 6 is very confusing to me*\
> We apologize for the confusion. Indeed, we mixed up the labels of Figure 6. We have now corrected them and improved the description in the main text. Furthermore, you suggest training models on the whole ImageNet training set and testing them on the ImageNet validation set without "trivials" and "impossibles". We are glad to report that we already implemented our subsampling analysis in the exact manner you describe. There was a typo in section 3.3: previously we wrote "[...] from the ImageNet training dataset" which we have corrected to "[...] from the ImageNet validation dataset". Sorry for the confusion; we hope that our corrections resolve the issues.
>
> *It is also unclear how surprising we should be towards the consistency distribution, is this a result of an exponential distribution of the general “identification” difficulty (most images are simple, then less and less are more difficult*?\
> Great suggestion! Previously in figure 2, we used the assumption that the image difficulty is uniform across all images (for the binomial baseline). Due to your suggestion we now modeled the image difficulty with an exponential decay (many images are simple, then less and less images are more difficult). We added a figure and a discussion of this approach in section A.8 of the appendix, which we also reference in section 3.1. Interestingly, the resulting distribution is still far away from our observed DDD distribution in figure 2. Thus, DDD does not simply result from an exponential decay of image difficulty. Thanks for this valuable suggestion!

---

> > ### Comment · Reviewer_rhYo · 2021-11-20
> > **response**
> >
> > Thanks for your response.
> >
> > I am looking forward to the upcoming data and analyses from the experiment.
> >
> > The new result addressing the exponential assumption is also interesting, though I should mention it’s different from what I imagined. I would not think that the current exponential distribution for the bar “None” should approach 0. I wonder whether that would change your current result.
> >
> > Thanks to your corrections on Figure 6. After looking at this new figure, I am now wondering whether the consistency between the models with different random initializations is too low to make the comparison on these images useful. After correcting the cross-model consistency using the within-model consistency, it still seems to me that the models are very similar to each other, am I interpreting the data correctly? Would you need to retrain these models on training set only with in-between images to make them more differentiated?

---

> > > ### Author Response · Authors · 2021-11-22
> > > **Author response to reviewer rhYo**
> > >
> > > Dear rhYo,
> > >
> > > thank you very much for your comment. First, please find the result of our experiment as general comment above (https://openreview.net/forum?id=C_vsGwEIjAr&noteId=QTo4J6xgBVk).
> > >
> > > *I would not think that the current exponential distribution for the bar “None” should approach 0. I wonder whether that would change your current result*.\
> > > Thanks for clarifying this. We changed the parameters of the exponential distribution so that the "None" bar does not approach zero (see Figure 9). However, this does not change the previous conclusion that DDD can not be fully explained by a decreasing exponential distribution (especially the occurrence of many images for which no model is correct). Does this address your concern?
> > >
> > > *It still seems to me that the models are very similar to each other, am I interpreting the data correctly? Would you need to retrain these models on training set only with in-between images to make them more differentiated*?\
> > > Thank you for your question. You wrote that "After correcting the cross-model consistency using the within-model consistency" the models still seem to be very similar. We are afraid that we do not quite understand what you are referring to. If "correcting the cross-model consistency using the within-model consistency" is your main concern, we would kindly ask you to elaborate on this. Nevertheless, we understand that you are concerned that the models seem to be very similar even after subsampling on the "in-betweens". We will therefore address this issue in the following paragraph:
> > >
> > > We find that the error consistencies are rather high, when evaluated on the complete ImageNet validation set. This is in line with previous results (see Geirhos 2020 and 2021). However, if we remove the "trivial" and "impossible" images, we see that the error consistency becomes close to zero (ResNet-18 variants) or zero (SOTA models). This means that models make truly independent decisions (see sec. 1.1 in Geirhos 2020). Independent decisions allow for the conclusion that model behavior is dissimilar. Furthermore, we would like to highlight that the small error consistencies resulting from the subsampling analysis are in stark contrast to results reported in the above previous work. Regarding your second question: since we already find very dissimilar models, we would argue that retraining the models on only the "in-between" images is not necessary.
> > >
> > > We hope that this clarified that from our point of view the subsampled models are not very similar.

---

> > > > ### Comment · Reviewer_rhYo · 2021-11-30
> > > > **Response**
> > > >
> > > > Thanks for your response. The new results from the follow-up experiment are interesting, as they show that there seem to be multiple factors for the difficult images being difficult. One question I have about the results is whether one difficult image would only have one "key-feature" or it could have multiple features, meaning whether the majority of the difficult images only get one "different" from usual answers in one of these questions but not from the others.
> > > >
> > > > About the similarity question I had about the models with the in-between images, my point about the correction is about dividing the off-diagonal value (i, j) using the square root of the multiple of the both diagonal values (i, i) and (j, j). From the current results, it seems that the corrected similarity is not low (like the Fig. 6 a, right table, the block from the different initialization to the different data loader). If you take 0.29 as the stability across different random seeds and treat that as an estimation for the diagonal values for the right table in Fig. 6b, then the similarities in that table are not that low either, although I agree that there are indeed several items that are close to 0. But we don't know the actual diagonal values for these items either. Therefore, I think retraining these models only on the in-between images seems to be an interesting next step to me to tell whether they are indeed different or not.
> > > >
> > > > After the revision and the new results, I am more willing to have this paper accepted, but I feel score 8 might be too high from my point of view (if there is a score 7, I would raise my score to it). So I will keep my current score.

---

> > > > > ### Author Response · Authors · 2021-12-01
> > > > > **Thank you for the discussion**
> > > > >
> > > > > Dear rhYo,
> > > > >
> > > > > thank you very much for your response. We are unsure what you mean with difficult images. In the following preliminary analysis we assume that images you refer to as difficult are those on which observers indicate that either there was no clear main category, the object presentation was odd, or both. We find that for 14% of answers, the main category was not clear and the item presentation was found to be odd at the same time. In contrast, observers indicated that there was a clear main category but the item presentation was odd in 19% of answers. Furthermore, the main category was not clear but the item presentation was found to be normal in 22% of answers. Taken together, it seems that more often, only a single factor was found to be "unnormal" for a given image.
> > > > >
> > > > > Regarding your second concern, why should we need to normalize the off diagonal with the diagonal elements? Cross model error consistency has a meaning on it's own. If ―for the SOTA models― error consistency is zero, the models give independent decisions regardless off the within consistency. This is in stark contrast to existing work where SOTA models on plain ImageNet are very similar and high cross-model error consistencies are reported (see for example: Geirhos et al., Oral at NeurIPS this year).
> > > > >
> > > > > We hope that we could clarify that the low cross-model consistencies already allow for the conclusion that the models are different. We will make sure ―together with the comment of MrqK― to address this in the final version.

---

> > > ### Author Response · Authors · 2021-11-26
> > > **Your input is highly appreciated**
> > >
> > > Dear Reviewer rhYo,
> > >
> > > recently we added two responses, one which includes the results of the new experiment ((https://openreview.net/forum?id=C_vsGwEIjAr&noteId=QTo4J6xgBVk) and another one directly related to your last comment (see below). Did you already have had a chance to look at them? Since the rebuttal deadline is quickly approaching, we would like to make sure that we have addressed your concerns.

---

### Official Review · Reviewer_oQTc · 2021-11-02

**Correctness:** 4
**Technical Novelty And Significance:** 4
**Empirical Novelty And Significance:** 4
**Recommendation:** 8
**Confidence:** 3

**Main Review:**


### Strengths
- 13 variants to ResNet-18 as well as 11 SOTA models where analyzed on
  ImageNet, CIFAR-100 and a gaussian toy dataset
- the writing is very engaging and overall very good
- the paper is very well structured, and offers many control experiments in the
  supplement
- the experiments are well documented for reproducibility

### Weaknesses

- minor: in A.4, the last paragraph is a copy from the first paragraph and footnote 6/7 are the same

I could not find any obvious weaknesses in this paper.
I think the paper is very well written, and the empirical experiments are
conducted thoroughly and without any obvious flaws.
The insights are very significant and novel, and I feel this paper is a very important contribution.


**Summary Of The Paper:**

The authors investigate why networks with highly different architectures and
objectives seem to produce similar decision boundaries.
By analyzing the model behavior of each sample of the validation set during
training, they find that over half of the samples of the ImageNet validation
set are either trivial or impossible for almost all analyzed models, which they
name "dichotomous data difficulty". They then show that the model agreement is
almost only caused by these trivial and impossible samples, by measuring the
agreement over all models with and without the trivial examples.
Finally, they show through human trial that humans can with a relatively high
accuracy (~81.36%) predict which samples are easier for neural networks to
predict, showing that there is a level of difficulty to the image samples.

**Summary Of The Review:**

Unless I have not missed anything very obvious, this submission seems very
polished, significant and novel.
Therefore, I recommend 8: accept, good paper.

---

> ### Author Response · Authors · 2021-11-15
> **Author response to reviewer oQTc**
>
> Dear reviewer oQTc, \
> thank you very much for your review, we appreciate the positive assessment of our work. Thanks for pointing out an issue in A.4 (new A.5), which is now fixed.

---

### Official Review · Reviewer_MrqK · 2021-11-04

**Correctness:** 4
**Technical Novelty And Significance:** 3
**Empirical Novelty And Significance:** 3
**Recommendation:** 6
**Confidence:** 5

**Main Review:**

**Strengths**

- The experiments on error consistencies across (i) models trained with different hyperparameters (figure 3) and (ii) state-of-the-art models (figure 2 and figure 5) are thorough and clearly showcase the effect of dichotomous data difficulty on model predictions.

- The psychophysical experiment results in section 3.4 are quite surprising, and suggest that humans and CNNs share a similar notion of image difficulty (albeit at a coarse level).

- The experiment on dataset subsampling is insightful, as it suggests that that the effect of model architecture and task complexity can be amplified in the absence of DDD / dataset issues. The drop in error consistency is quite drastic, so it would be good to have the individual model accuracies reported in the paper as well.

**Weaknesses**

- Novelty vis-a-vis prior work.
    - Papers on example difficulty [R1-R4] (and references therein) already show that a non-trivial fraction of data points are systematically "easy" and "hard" across different models etc. In this context, DDD is simply a different way to frame previously known results.

    - The purpose of Figure 4 is unclear. The takeways reported from figure 4 are already known observations from previous works. [R5, R6] discusses learning order and [R4] discusses training time as a proxy for example difficulty.

    - Hard examples can also originate from "noisy" datasets that contain data points with label errors (such as imagenet) .The analysis in this paper does not take into account the effect of label errors on example difficulty (the impossible set of points). Label errors can erroneously inflate the metrics used to describe DDD in the paper.

    - "...all models end up with a similar decision boundary". This statement is incorrect. While models with similar decision boundaries will make similar errors, it is not necessary that models that make similar errors have similar decision boundaries. There is a separate line of work [R7, R8] that study feature learning and decision boundaries despite differing inductive biases of different architectures.

- Scope of empirical analysis limited. While all experiments in the paper serve to identify DDD, it is not clear if this is a dataset "issue" per se. Is it reasonable or even possible to curate large-scale datasets that do not have DDD? Answering these questions requires additional analysis on what it means for examples to be "trivial" or "impossible". For example, are all "trivial" examples within a class similar in some aspect? are "in-between" images contributing the most to the model's performance? Experiments or preliminary analysis in this direction would have made the paper (and its conclusions) stronger.

---

**References**

[R1] Hacohen, G., Choshen, L. and Weinshall, D., 2020, November. Let’s agree to agree: Neural networks share classification order on real datasets. In International Conference on Machine Learning (pp. 3950-3960). PMLR.

[R2] Agarwal, C., D'souza, D. and Hooker, S., 2020. Estimating example difficulty using variance of gradients. arXiv preprint arXiv:2008.11600.

[R3] Baldock, R.J., Maennel, H. and Neyshabur, B., 2021. Deep Learning Through the Lens of Example Difficulty. arXiv preprint arXiv:2106.09647.

[R4] Mangalam, K. and Prabhu, V.U., 2019. Do deep neural networks learn shallow learnable examples first?.

[R5] Toneva, M., Sordoni, A., Combes, R.T.D., Trischler, A., Bengio, Y. and Gordon, G.J., 2018. An empirical study of example forgetting during deep neural network learning. arXiv preprint arXiv:1812.05159.

[R6] Kalimeris, D., Kaplun, G., Nakkiran, P., Edelman, B., Yang, T., Barak, B. and Zhang, H., 2019. Sgd on neural networks learns functions of increasing complexity. Advances in Neural Information Processing Systems, 32, pp.3496-3506.

[R7] Wang, L., Hu, L., Gu, J., Wu, Y., Hu, Z., He, K. and Hopcroft, J., 2018. Towards understanding learning representations: To what extent do different neural networks learn the same representation. arXiv preprint arXiv:1810.11750.

[R8] Shah, H., Tamuly, K., Raghunathan, A., Jain, P. and Netrapalli, P., 2020. The pitfalls of simplicity bias in neural networks. arXiv preprint arXiv:2006.07710.

**Summary Of The Paper:**

This paper analyzes the effect of dichotomous dataset difficulty (DDD) on model predictions; it has three key findings. First, it shows that a large fraction of imagenet images are either trivial (most models classify such images correctly) or impossible (most models classify such images correctly) for several state-of-the-art models. Second, it shows that differences between model predictions get pronounced when models are trained on "in-between" images only (i.e., in the absence of DDD). Third, it suggests that humans and CNNs have similar notions of image difficulty by showing that humans are highly accurate at predicting which images are difficult for CNNs

**Summary Of The Review:**

Overall, I think the weaknesses of this paper outweigh its strengths. While the experiments are thorough and the human study results are quite interesting, I am primarily concerned about the novelty of the empirical findings (+ missing related work) and the limited scope/breadth of experiments (e.e. analysis of easy and hard examples).

---

> ### Author Response · Authors · 2021-11-15
> **Author response to reviewer MrqK [I/II]**
>
> Dear reviewer MrqK,
>  thank you very much for your comments and the excellent literature recommendations. These will clearly improve our paper.
>
> *What does it mean for examples to be "trivial" or "impossible"*?\
> This is an excellent suggestion which is shared with the other reviewers. We address this question with the follow-up psychophysical experiment outlined above. We will get back to you with the results in a few days, after data collection and analyses are complete.
>
>
> *The analysis in this paper does not take into account the effect of label errors*\
> We agree that label errors these are an important factor to consider when analysing network decisions---in fact, we already discussed (and analyzed) the influence of label errors in various locations: Abstract, Introduction, Method and Discussion. Label errors are also plotted in Figure 2, which shows that their influence on the group of "impossible" images is only minor. We revisit this issue in the second paragraph of the discussion. We have now made sure to make this link clearer. Does this resolve the issue?
>
>
> *Can we curate DDD-free datasets*?\
>  Your comment highlights a very important point. It is possible to curate DDD-free datasets. For this purpose, one would simply have to train multiple models on the same data. Afterwards, the trivial and impossible images would have to be removed. This is an easy to implement and effective approach to attain DDD-free datasets. Since we think ML-practitioners could benefit from DDD-free datasets, we added this to the discussion section.
>
> *It is already known that many examples are systematically easy / hard. How does DDD differ from previously known results*?\
> First of all, thanks for pointing out a number of related references that we failed to cite in our initial submission. We have now added an additional paragraph to section 1.1 called "example difficulty" to address this, where we discuss our motivation in the context of existing work on example difficulty. As you pointed out, a number of papers investigated easy and hard examples---e.g. references R2--R4 for MNIST/CIFAR, and R1 for ImageNet. Thus, where is the difference to DDD? While it was known that models often make similar errors /learn examples in the same order, it was unclear whether this was caused by models or by data. We here present evidence that there is a major problem with popular image datasets that has a decisive influence on network decisions. This leads to the true model differences being masked by many "trivial" and "impossible" images. Furthermore, we also think that this issue is not known to the full extent by the ML community. Many papers investigate model differences on the standard ImageNet validation set and do not take into account that the model similarity is partially driven by DDD. We don't mean to argue that this reduces the contributions of these papers in any way---in contrast, we believe our findings shed light on previous results since they disentangle the influence of model inductive bias from example difficulty.
>
>
> *Figure 4 is already known from previous work*\
> Agreed, the key takeaways from Figure 4 are already known from previous works. We decided to include Figure 4 since we believe it might be hard to understand Figure 5 without first understanding Figure 4; and we found Figure 4 a nice "at a glance" visualisation of 4.5M model decisions (50K test set examples across 90 epochs). We now explicitly link R4, R5 and R6 when discussing Figure 4 in order to give credit to those who have investigated model errors over (training) time before. It now reads as follows: "The key takeaways from Figure 4 are already known from previous works investigating model errors over training time (Toneva et al., 2018; Mangalam and Prabhu, 2019; Kalimeris et al., 2019)---we do not intend to claim any conceptual novelty in this regard, Figure 4 simply intends to visualise these intriguing patterns clearly."

---

> > ### Author Response · Authors · 2021-11-15
> > **Author response to reviewer MrqK [II/II]**
> >
> > *Rephrasing ...all models end up with a similar decision boundary*\
> > Good point, our writing needs to be more precise regarding this. We now refrain from making statements about similar decision boundaries, since error consistency does not consider the underlying geometry of high-dimensional decision boundaries. Instead, error consistency measures whether models make similar decisions(which may be just as relevant for many practical purposes). We therefore changed the following statements:
> > "[...] all models end up with a similar decision boundary" to "[...] all models end up making similar decisions" (Abstract) and "[...] does not change the decision boundary significantly" to "[...] does not change the decisions significantly" (Section 3.2).
> >
> > *It would be good to have the individual model accuracies reported in the paper as well*\
> > In the previous version of our manuscript we only reported the accuracies of the ResNet-18 models. We have updated our manuscript to also include the individual accuracies of the SOTA models (see Figure 11).

---

> > ### Comment · Reviewer_MrqK · 2021-11-22
> > **Response to rebuttal**
> >
> > Thanks for the response. The rebuttal addresses my concerns about label errors, trivial vs. in-between vs. impossible images (the new experiment design addresses this), and contextualising results w.r.t. previous works. However, I still have questions/concerns about (i) past work on example difficulty vs. DDD phenomenon and (ii) validity and usefulness of DDD-free datasets:
> >
> > > As you pointed out, a number of papers investigated easy and hard examples---e.g. references R2--R4 for MNIST/CIFAR, and R1 for ImageNet. Thus, where is the difference to DDD? While it was known that models often make similar errors /learn examples in the same order, it was unclear whether this was caused by models or by data.
> >
> > I am not entirely convinced by this argument. Based on my understanding of  prior work on easy/hard examples, the results show that models _with different architectures_ have a similar "learning order". That is, the examples that are easy  (learned first) and hard (learned towards the end) are similar for different architecture. This strongly suggests that data has an over-sized impact on model decisions, independent of which model architecture is used.
> >
> > > Afterwards, the trivial and impossible images would have to be removed. This is an easy to implement and effective approach to attain DDD-free datasets.
> >
> > If you take a dataset, train some models, remove the "trivial" and "impossible" examples, and retrain a new model on the DDD-free dataset, it can still exhibit DDD w.r.t. the new (retrained) models? Isn't it DDD-free only w.r.t. the models trained on the "original" dataset? Also, I have concerns about actionable benefits of DDD-free datasets. What insights can practitioners gain from DDD-free datasets that cannot be obtained by looking at subpopulation performance or OOD datasets? DDD-free datasets will amplify the differences that are already present but what concrete implications does this have for practitioners?

---

> > > ### Author Response · Authors · 2021-11-22
> > > **Author response to reviewer MrqK**
> > >
> > > Dear MrqK,
> > >
> > > thanks for clarifying your reasoning. First, please find the result of our experiment as general comment above (https://openreview.net/forum?id=C_vsGwEIjAr&noteId=QTo4J6xgBVk).
> > >
> > > *The examples that are easy (learned first) and hard (learned towards the end) are similar for different architecture. This suggests that data has an over-sized impact on model decisions, independent of which model architecture is used*. \
> > > True! We agree that it was previously known that the data have a strong impact on model decisions. However, this is different from our main conclusion: model similarity is highly driven by the dichotomous nature of image difficulties. While the existence of very hard and very easy images had been shown before (as you rightly highlight with the references you provided and which we no include in our literature section), their effect on model similarity had not been well understood.
> > >
> > > Most importantly, we argue that the issue of DDD is not recognized at all in the machine learning research community: very recent and high impact publications investigating model similarity do not take into account the effects of "trivials" and "impossibles" (see for example: Geirhos 202O, Oral at NeurIPS or  Mehrer 2020, Nature Communications).  We want to make sure that we communicate our main conclusion as clearly as possible.
> > >
> > > Previously we wrote: \
> > > "A number of papers have investigated what makes images easy or difficult—e.g. Agarwal et al. (2020); Baldock et al. (2021); Mangalam and Prabhu (2019); Paul et al. (2021) for MNIST/CIFAR, and e.g. Hacohen et al. (2020) for ImageNet. Furthermore, However, while it was previously known that models often make similar errors—and often learn examples in the same order, see e.g. Toneva et al. (2018); Kalimeris et al. (2019)—it is still unclear whether this is caused by models or by data, which we attempt to disentangle here."
> > >
> > > This passage now reads:\
> > > "A number of papers have investigated what makes images easy or difficult—e.g. Agarwal et al. (2020); Baldock et al. (2021); Mangalam and Prabhu (2019); Paul et al. (2021) for MNIST/CIFAR, and e.g. Hacohen et al. (2020) for ImageNet. Additionally, it was previously known that models often make similar errors--and often learn examples in the same order, see e.g. Tonevaet al. (2018); Kalimeris et al. (2019).\
> > > To summarise: it was clear that there are easier and harder images and that models often make similar errors. In contrast and most importantly, the relationship between these two findings was not recognised. We here show for the first time that the true model similarities are masked by the dichotomous data difficulty."
> > >
> > > *If you take a dataset, train some models, remove the "trivial" and "impossible" examples, and retrain a new model on the DDD-free dataset, it can still exhibit DDD w.r.t. the new (retrained) models? Isn't it DDD-free only w.r.t. the models trained on the "original" dataset*? \
> > > In principle you are correct. If we were to remove the "trivial" and "impossible" examples and retrain models on only the "in-between" images, it is highly likely that new "trivial" and "impossible" examples will emerge. After all, which images are "trivial" and "impossible" depends on the models and the data set used. However, it remains an open question whether this training procedure could potentially improve the robustness and accuracy of the models. This is a very interesting hypothesis which should be carefully investigated in future studies. Due to the already very extensive nature of our present manuscript, we think that a proper execution of this analysis is outside the scope of our current work.
> > >
> > > *What insights can practitioners gain from DDD-free datasets that cannot be obtained by looking at subpopulation performance or OOD datasets*? \
> > > For practitioners, DDD has an impact. Imagine that you train two (or more models). In an ideal world, you would like to see an error consistency of one, such that there is no difference between these two models as they make the same decisions. In the real world, DDD will be present in a subset of images for many datasets. Here, we do not argue against the effectiveness of subpopulation analysis. In contrast, we think that subpopulation analyses focusing on the "trivials", "in-betweens" and "impossibles" could help practitioners better understand what their models learn and don't learn, subsequently allowing them to adapt their training procedure. In a similar vein, it was very recently shown that models with uncorrelated errors "specialize in subdomains of the data, leading to higher ensemble performance" (Gontijo-Lopes, 2021 also under review at ICLR 2022). Therefore, focusing on the "in-between" images, where model decisions are almost completely uncorrelated, would yield better ensemble models. Without a doubt, this is of relevance for practitioners.

---

> > > > ### Comment · Reviewer_MrqK · 2021-11-23
> > > > **Response**
> > > >
> > > > *Model similarity*: I now understand your argument about DDD vis-a-vis model similarity:
> > > > - DDD / example difficulty needs to be taken into account while measuring model similarity and current approaches (eg. representation similarity methods) do not take this into account.
> > > > - DDD-free datasets (i.e. identifying in-between images) can reveal differences between _trained_ models more clearly. As you mentioned in the response above, it can also be used with subpopulation metadata / group information in a complementary way.
> > > > - DDD-free datasets can be used to potentially improve ensemble performance (e.g. ensembling models with decisions that are least correlated on the set of in-between images).
> > > >
> > > > *Psychophysical experiments*: The results (and design) of the new follow-up experiment are interesting! Along with the first human study, I think these experiments adequately address my initial concern about limited analysis of trivial/in-between/impossible. These experiments add a new dimension to the example difficulty research direction, which has so far only focused on summary statistics obtained while training the model.
> > > >
> > > > *Writing*: To me, the most insightful aspect of this paper are the psychophysical experiments. However, most of the paper is about identifying DDD / analysing example difficulty (fig 3,4,5). So, I think the paper will be more interesting to the community if the main paper has more emphasis on (i) results of both psychophysical experiments and (ii) model similarity results with DDD-free datasets, and less emphasis on (i) robustness of DDD results w.r.t. hyper-parameters, (ii) example difficulty analysis during training and (iii) discussion. However, this probably requires significant restructuring..
> > > >
> > > > ---
> > > >
> > > > I am increasing my score to 6:
> > > >
> > > > (+) My primary concerns about limited analysis of trivial/impossible images, connections to work on example difficulty and DDD-free datasets have been addressed.
> > > >
> > > > (-) I think the paper (as of now) overemphasizes results that are not significantly novel. However, this is minor and hopefully can be fixed (see above).

---

> > > > > ### Author Response · Authors · 2021-12-01
> > > > > **Thank you ― we appreciate the discussion!**
> > > > >
> > > > > Dear MrqK,
> > > > >
> > > > > thank you very much for all the input during the discussion phase. The new analyses improved our paper. Furthermore, the discussions about image difficulty vs. model similarity is important for us. We agree that the model similarity aspect together with our psychophysical experiments are very interesting for the community. We will make sure that our final version emphasizes novel aspects. Thank you very much for increasing your score.

---

### Author Response · Authors · 2021-11-11
**Follow-up experiment --- we would highly appreciate your comments**

Dear Reviewers,

Thank you very much for your insightful comments. We will make sure to address them to the best of our ability over the next few days. We noticed that reviewers MrqK, rhYo and 3xZg suggested having a closer look at differences between "trivial", "impossible" and "in-between" images. This is an excellent suggestion. For this purpose, we will conduct a follow-up experiment  with naïve human observers on a random subset of "trivial", "impossible" and "in-between" images. Therefore, we would like to ask you for any feedback/suggestions you may have regarding the experiment until November 14.

Specifically, we propose the following experimental procedure:

"Imagine you have a job at an image database company. You constantly get new images and have to do an initial assessment for your boss.  Please answer the following questions for every image."

1. How many objects belonging to different categories are in the image (e.g. three dogs are still one category: dog. But two dogs and one cat are two categories: dog and cat)?
2. Is there a main object in this image? (Yes, maybe, no)
3. Is the presentation of the objects unusual in any manner (e.g. orientation, location, size, viewpoint)? (No, slightly, very)

We piloted the above experiment yesterday. In the debriefing, subjects reported being confused by having to differentiate multiple aspects related to the presentation of the objects such as orientation and viewpoint. We therefore grouped a number of questions regarding these aspects into question number three.
Due to your feedback and the results from our pilot experiment, we think that the most interesting aspect is to disentangle whether impossible images consist of either multiple objects or "oddly presented" objects (or something else entirely), which we think is covered by our questions. However, we would like to use OpenReview's unique opportunities for author-reviewer interaction to ask you for any feedback you may have regarding this---do these aspects cover the hypotheses you had in mind? Is there any other aspect that we should include in this follow-up experiment?

---

> ### Comment · Reviewer_MrqK · 2021-11-13
> **Questions about follow-up experiment**
>
> Thanks for initiating a discussion about the follow-up experiment. This seems like a reasonable way to better characterize the differences in trivial/in-between/impossible images from a "human" perspective. I have a few questions about the experiment:
>
> - Could you clarify what you mean "naive" human observers? Does it mean that the observers have limited knowledge about image datasets / machine learning in general?
>
> - "How many objects...dog and cat": Is the list of categories provided to the observers? If not, observers might have different notions of categories (e.g. dog/cat/fish or animal/object)..
>
> - "Is there a main object in this image?": Just to clarify, if there multiple dogs for example, does this mean there is no main object? Is still seems like the dominant category. I think it'd be less confusing if the question is "is there a main category in the image?"
>
> Based on the results, it might be interesting to check how model accuracy varies with (i) number of categories in images and (ii) whether or not there is a main category.

---

> > ### Author Response · Authors · 2021-11-15
> > **Thank you very much for your input**
> >
> > Dear reviewer MrqK,
> > thank you very much for your comments about the follow-up experiment.
> >
> > *Could you clarify what you mean "naive" human observers*?\
> > In psychophysics, the term "naive" is usually used to refer to observers who are unaware of the specific goals of a study. We plan to use different levels of experience: some observers do not have experience with (DNN-) research at all, and some observers will be experts. This will allow us to see if humans agree on their judgements between the three image classes, and if this agreement differs based on their experience (see also next comment).
> >
> >
> > *How many objects...dog and cat": Is the list of categories provided to the observers? If not, observers might have different notions of categories (e.g. dog/cat/fish or animal/object)*..\
> > You are right, it is possible that observers might have different notions of categories. But wouldn't this be a finding on its own? What is interesting to us is the first response that comes to mind. This is called entry-level or basic-level category (Rosch 1988 and Geirhos 2018). If we provide categories we loose discriminatory power between observers since we wouldn't recognise if observers would naturally use different categories. We don't want to bias the observers. Of course, we will check the observer consistency after the experiment and we will ask the subjects regarding their notions of the categories in the debriefing. We hope that this clarifies our reasoning.
> >
> >
> > *Is there a main object in this image?: Just to clarify, if there multiple dogs for example, does this mean there is no main object? Is still seems like the dominant category. I think it'd be less confusing if the question is "is there a main category in the image*?\
> > This is an excellent suggestion! We agree, it is better to ask for a main object category. We will adapt our question accordingly.
> >
> > Thank you very much again for the time you spent reviewing. We are already looking forward to reporting our results.

---

### Author Response · Authors · 2021-11-22
**Follow-up experiment --- Results**

A central question that was raised by our reviewers (MrqK, rhYO and 3xZG) is: What makes "trivials", "impossibles" and "in-betweens" different? More specifically rhYo suggested: *The authors could provide more evidence to either prove or disprove whether "trivals", "in-betweens" and "impossibles" show differences in aspects like atypical object pose or position*?

We believe this is an important suggestion. In order to address this we conducted a follow-up psychophysical experiment, see section A.9. Our goal was to understand which factors are underlying the differences between "trivials", "in-betweens" and "impossibles".

For this purpose, we randomly chose 100 "trivial" (all networks give the correct response), 100 "impossible" (no network gives a correct response) and 100 "in-between" images. Each observer had to answer the following questions for each image: "How many objects belonging to different categories are in the image (e.g. three dogs are still one category: dog. But two dogs and one cat are two categories: dog and cat)?", "Is there a main category in the image? (no, maybe, yes)" and "Is the presentation of the objects unusual in any manner (e.g. orientation, location, size, viewpoint)? (no, slightly, very)". Our main results are summarised in figure 10. We show that the number of categories in the image, whether there is a main category or not, and the oddness of presentation all seem to contribute to whether an image belongs to the "trivial", "in-between" or "impossible" subsets.

MrqK asked *Based on the results, it might be interesting to check how model accuracy varies with (i) number of categories in images and (ii) whether or not there is a main category*. That's a great suggestion.

For (i), the Pearson correlation coefficient between the mean number of categories over all observers for each image and the respective mean model accuracy for the same image is -0.37.

For (ii), we found the following mean model accuracies for each of the answers:\
Main category "yes":        0.64\
Main category "maybe":   0.44\
Main category "no":          0.37

Odd presentation "no":        0.61\
Odd presentation "slightly": 0.55\
Odd presentation "very":     0.34


Futhermore, reviewer MrqK asked: *Are all "trivial" examples within a class similar in some aspect*? Reviewer 3xZg asked: *Are the subsets clustered around certain classes*? \
To answer these question we grouped all 1000 ImageNet classes into 11 superclasses based on the WordNet hierarchy (section A.10 in the appendix). In figure 11 and figure 29 we can see that the errors are not uniformly distributed between the superclasses. While "trivial" examples are present in all superclasses, there are easier and harder superclasses, with "birds" being particularly easy and "implements, containers, misc. objects" being the hardest. Upon visual inspection, this makes sense: many bird photographs are professional photographs of a single bird against an uncluttered background (easy), while many images from the "images, containers, misc. objects" superclass contain multiple objects at once in a rather cluttered setting.


We think these analyses provide important insight into what makes "trivials", "impossibles" and "in-betweens" different. We would like to thank our reviewers for their suggestions which definitely make our paper stronger.

---

### Author Response · Authors · 2021-11-22
**Author summary of rebuttal process**

We would like to thank all reviewers for their valuable feedback and we very much appreciate their assessment of our work as **thorough and solid** (MrqK, oQTc, rhYo) with **fascinating experiments** that **point to many potential challenges in today's evaluation of image classification models** (3xZg). MrqK describes the results of our experiment as **insightful** and **quite surprising**; oQTc points out that **the insights are very significant and novel**, and that the **paper is a very important contribution**. Furthermore, we were happy to hear that the writing is **very engaging and clear** (oQTc, 3xZg).

We addressed the individual concerns of each of the reviewers below. Here is a summary of their main concerns and how we addressed them:

- Based on excellent feedback by reviewers MrqK, rhYO and 3xZG, we conducted a follow-up psychophysical experiment to find differences between "impossible", "in-between" and "trivial" images. We show that the number of categories, whether or not there is a clear main category, and how odd the presentation of objects is, seem to be decisive factors for differences between "impossible", "in-between" and "trivial" images. See section A.9.
- Following suggestions by reviewers MrqK and 3xZG, we analysed the proportions of images from the three image subsets belonging to each of the ImageNet superclasses. Here we find that the image subsets are not equally distributed over the superclasses. For example, the superclass "Birds" is dominated by "trivial" images, see section A.10.
- In response to reviewer MrqK, we added a section about existing example difficulty literature (section 1.1). Additionally and in order to underline the significance of our work, we discussed the implications of DDD-free datasets and their relevance for practitioners.
- Reviewer rhYO suggested to check whether an exponential distribution of the classification difficulty (most images are simple, then less and less are more difficult) could explain DDD. Our simulation in section A.8 shows that an exponential difficulty distribution can not explain DDD.
- Finally, in a re-analysis of existing data for reviewer 3xZG, we show evidence that observers agree with their classifications on the "in-between" images, see section A.11.

We again want to thank the reviewers for their time and for actively taking part in the discussion process. The suggestions of the reviewers have without a doubt improved the breadth and quality of our submission.

---

### Comment · Area_Chair_MyNf · 2021-11-29
**Using OOD data for discerning models**

Dear Authors,

I wanted to ask one follow-up question on using datasets without the DDD property for evaluating models. Measuring performance on out-of-distribution data is a popular way to evaluate models. In particular, it can differentiate models that exhibit small differences on the in-domain data. Hence, using OOD data seems to be an alternative to gathering datasets without the DDD property. Could you please comment on the relative strengths and weaknesses of using OOD data vs building datasets without the DDD property?

Thank you,

AC

---

> ### Author Response · Authors · 2021-12-01
> **Response to AC ― Clarifying the relationship between OOD and DDD**
>
> Dear AC,
>
> Thanks for this interesting question, we appreciate your interest!
>
> Indeed, evaluating models on out-of-distribution (OOD) datasets is an important way to differentiate between models that generalise well and those that don't (in terms of accuracy). As far as we are aware, however, even evaluating models on OOD data does not help much in terms of going beyond high model-to-model error consistency: Different models (e.g. Squeezenet vs. ResNet-152 vs. Inception-v3) show high error consistency even when evaluated on OOD data, according to Figure 3 of Geirhos et al. (2020, https://arxiv.org/pdf/2006.16736.pdf). Here, CNN-to-CNN consistency is at .62, .48 and .67 depending on the OOD dataset, which is closer to perfect consistency than it is to chance level. Therefore, it appears that while OOD data can distinguish models in terms of overall accuracy, it is insufficient in terms of revealing deeper differences overshadowed by DDD. (It remains an interesting open question whether e.g. training on a DDD-free dataset would affect OOD accuracies in any way. Furthermore, OOD datasets might also exhibit DDD.) Looking forward, OOD testing as well as curating datasets without DDD are not mutually exclusive and should be employed in a combined fashion for a comprehensive understanding of model similarities and differences.
>
> We hope this addresses your question, and we will make sure to add a comment on the relationship between OOD and DDD for the camera ready version.

---

### Decision · Program_Chairs · 2022-01-20

**Decision:**

Accept (Poster)

**Comment:**

The key contribution of the paper is identifying that datasets have the so-called DDD property. In short, datasets are predominantly composed of examples that are either consistently trivial or challenging (often misclassified) for neural networks.

Reviewer MrqK pointed out that it is well known (and provided four references) that many examples are consistently very hard or very easy for neural networks. This is true. It is somewhat novel how the authors attribute this phenomenon to datasets. Here, I would like to note that I slightly disagree with the attribution of the phenomenon to dataset alone. While it is a property of datasets, it is not self-evident that deep nets trained with SGD have to learn these trivial examples. Attributing it either only to dataset or to model/optimization seems to be oversimplistic.

The second key issue of the paper is that it is somewhat inconclusive. Many datasets have the DDD property, but the Authors provide a somewhat unclear motivation for why it matters. In particular, the fact the two models make correlated errors on a dataset does not mean we cannot distinguish them. In fact, we have been able to distinguish models using IID and OOD datasets. They make correlated errors, but one makes, with a significant margin, less errors than the other. Having said that, I agree that we without the DDD property we would be able to more easily distinguish models. This is a useful perspective.

Reviewers appreciated added experiments that help better characterize what are these trivial and impossible examples.

Despite the issues with novelty and framing, I think it is a useful perspective and hopefully will encourage more research into understanding the interaction between data and training. It is my pleasure to recommend acceptance and thank you for submitting the paper.

In the camera ready, please: (a) describe much more clearly and openly relation to prior work; (b) bring to the main text more data from the psychophysical experiments; and (c) address any other remarks made by reviewers.